# MVGaussian: High-Fidelity text-to-3D Content Generation with Multi-View Guidance and Surface Densification

## Abstract

The field of text-to-3D content generation has made significant progress in generating realistic 3D objects, with existing methodologies like Score Distillation Sampling (SDS) offering promising guidance. However, these methods often encounter the *Janus* problem—multiface ambiguities due to imprecise guidance. Additionally, while recent advancements in 3D Gaussian splatting have shown its efficacy in representing 3D volumes, optimization of this representation remains largely unexplored. This paper introduces a unified framework for text-to-3D content generation that addresses these critical gaps. Our approach utilizes multi-view guidance to iteratively form the structure of the 3D model, progressively enhancing detail and accuracy. We also introduce a novel densification algorithm that aligns Gaussians close to the surface, optimizing the structural integrity and fidelity of the generated models. Extensive experiments validate our approach, demonstrating that it produces high-quality visual outputs with minimal time cost. Notably, our method achieves high-quality results within half an hour of training, offering a substantial efficiency gain over recent 3DGS-based methods such as GSGen ($\sim$2 hours) and LucidDreamer ($\sim$35 minutes), reducing training time by up to 2$\times$ while achieving comparable or better results. Project page: [mvgaussian.github.io](https://mvgaussian.github.io).

## 1 Introduction

Recent advancements in text-to-3D generation have opened new avenues for creating complex 3D content directly from textual descriptions. This capability is crucial as it provides a straightforward, intuitive means for creators across various industries like gaming, virtual reality, and film-making, enabling rapid prototyping and visualization without the need for advanced modeling software or specialized training.

In leveraging foundation models for image generation, recent works have used reconstruction methods like Neural Radiance Fields (NeRFs) (Mildenhall et al., 2020) and 3D Gaussian Splatting (3DGS) (Kerbl et al., 2023) to make significant strides in the field. These models typically utilize Score Distillation Sampling (SDS) (Poole et al., 2022) to train the NeRF or Gaussian splatting methods, allowing for the generation of consistent 3D representations suitable for high-quality rendering and mesh extraction.

Recent approaches (Chen et al., 2024; Liang et al., 2023; Wang et al., 2023b; Yi et al., 2024; Tang et al., 2024) have successfully generated 3D models, yet they face significant challenges that limit their practical applications. These challenges include the multi-face (or Janus) problem, in which models produce inconsistent appearances from different angles, lengthy training times, and a general lack of fine detail in the generated models. Furthermore, most methods suffer from issues related to the complexity of their components and hyperparameters. They require considerable computational resources and time to generate high-quality content, or they compromise on quality to achieve faster processing times due to the inherent trade-off between quality and speed.

To address these limitations, we propose a novel framework that enhances the text-to-3D content generation pipeline by integrating SDS with an efficient 3D Gaussian splatting representation. Our approach not only tackles the aforementioned issues but also significantly reduces the computational overhead and training time. Our contributions can be summarized as follows:

- We introduce a text-to-3D framework that combines 3D Gaussian Splatting with multi-view diffusion guidance and novel regularization terms, enabling more efficient optimization with reduced Janus artifacts.

- We propose a novel densification method by optimizing the placement and density of Gaussian elements that accelerate the generation process reducing the overall training time to $\sim 25$ minutes.

- Through rigorous experiments, we demonstrate that our method not only matches but often surpasses the quality of existing SDS-based approaches with shorter training time.

## 2 Related work

Recent advancements in text-to-3D synthesis are built on the foundations established by text-to-image generation, 3D representations, and techniques for lifting 2D images to 3D models. This section reviews significant contributions in these areas, highlighting their methodologies and addressing their limitations.

### 2.1 Text-to-image generation

Earlier works in text-to-image generation leveraged GANs to map sentences to realistic images Li et al. (2019). With the advent of diffusion models (Ho et al., 2020b; Song et al., 2020), the field of text-to-image generation has advanced significantly, accelerating progress in content generation. Stable Diffusion (Rombach et al., 2022a) has demonstrated the effectiveness of diffusion over latent spaces for producing high-quality conditioned generations, particularly for text-to-image tasks.

Methods such as DALL-E (Ramesh et al., 2021) and Imagen (Saharia et al., 2022) utilize text embeddings, such as CLIP (Radford et al., 2021), to jointly train text and image encoders and decoders. These models are trained on large-scale datasets, such as LAION (Schuhmann et al., 2022), enabling zero-shot image generation. The method proposed by Nichol et al. (2021) explores two approaches—CLIP guidance and classifier-free guidance—and demonstrates that the latter is preferred in human evaluations.

Recent works have extended these approaches to multilingual image generation Ye et al. (2024). Additionally, text-to-image diffusion models have been further explored in image editing applications, such as Instruct Pix-to-Pix (Brooks et al., 2023), leveraging advancements in image inversion techniques (Mokady et al., 2023; Gal et al., 2022). Controlled image generation has also been a focus in methods like ControlNet (Zhang et al., 2023), DreamBooth (Ruiz et al., 2023), and InteractDiffusion (Hoe et al., 2024). Beyond image generation, these models encode extensive semantic knowledge, making them effective for zero-shot classification tasks, as demonstrated by Clark & Jaini (2024).

### 2.2 3D Representations

Recent advancements in 3D volumetric rendering have focused on using a shallow neural network that learns to represent complex scenes as Neural Radiance Fields (NeRF) (Mildenhall et al., 2020). This network predicts $RGB\sigma$ values at a given point as viewed from a certain direction. The optimization is performed using a ray-marching setup, which has proven effective for novel view synthesis even with sparse views. This approach has also been extended to temporal scenes (Mildenhall et al., 2020; Cao & Johnson, 2023). A vast body of work has rapidly emerged that builds upon these methods by exploring their various attributes. NeRFs have been investigated from different perspectives, including the use of sparse views Guangcong et al. (2023), generating NeRFs from unknown camera parameters (Lin et al., 2021), and reconstructing refractive surfaces (Guo et al., 2022). Additionally, they have been extended to be queried using language models (Kerr et al., 2023) and have been employed for 4D or higher-dimensional representations (Fridovich-Keil et al., 2023).

Shifting from implicit to explicit representations, Kerbl et al. (2023) introduced 3D Gaussians with a differentiable rasterization technique for faster, and real-time rendering. This method optimizes Gaussian parameters like scale, rotation, opacity, and color, and includes gradient-based schemes for managing Gaussians in a scene. Due to its speed and efficiency, this approach has largely replaced NeRFs in many applications such

as 3D content generation (Chen et al., 2024; Yi et al., 2024; Liang et al., 2023; Liu et al., 2024), SLAM (Keetha et al., 2024; Matsuki et al., 2024; Pham et al., 2024), and semantic scene understanding (Qin et al., 2024; Peng et al., 2025).

### 2.3 Lifting to 3D

Building on previous methods, Poole et al. (2022) introduced a novel approach for generating 3D models by leveraging text-to-image models. Their method employs a pre-trained diffusion model to distill multi-view information into NeRF models. The core of this approach is the Score Distillation Sampling (SDS) technique, which optimizes the NeRF model while omitting the U-Net Jacobian term. However, due to the lack of 3D awareness in the Stable Diffusion model, this method suffers from limitations such as blurry renderings and the Janus problem.

To address these issues, Wang et al. (2023a) introduced voxel radiance fields and Score Jacobian Chaining, improving image quality while still facing challenges like multiview inconsistancy and mode collapse. Prolific Dreamer (Wang et al., 2023b) further enhanced visual quality, diversity, and robustness by introducing Variational Score Distillation. DreamGaussian (Tang et al., 2024) replaced the NeRF-based representation with 3D Gaussian Splatting (3DGS), enabling faster text-to-3D and image-to-3D generation. However, it still encounters issues such as the Janus problem (Poole et al., 2022), poor mesh quality, and a lack of fine details. To improve aesthetics, Mathur et al. (2023) introduced an aesthetic score function, leveraging reinforcement-based techniques in conjunction with SDS. Similarly, Ye et al. (2025) proposed a novel reward function based on consistency, user preferences, fidelity, and alignment, using human-annotated data to train text-to-3D models. They optimize generation quality by combining a pre-trained multi-view score model with a diffusion model.

Beyond diffusion-based approaches, other 3D generation methods have been explored. For instance, Point-E (Nichol et al., 2022) generates text-to-point clouds, while DMTet (Shen et al., 2021) employs a differentiable tetrahedral representation for 3D reconstruction. Several subsequent methods, including GSGen (Chen et al., 2024), LucidDreamer (Liang et al., 2023), and GaussianDreamer (Yi et al., 2024), incorporated Point-E for initialization. However, Point-E struggles to generalize to complex prompts, limiting its effectiveness. In contrast, Chen et al. (2023a) introduced a unique approach that disentangles geometry and appearance for high-quality 3D generation, leveraging a DMTet-based hybrid surface representation (Shen et al., 2021).

Recent works draw ideas from multiple different works discussed previously and find unique solutions. GaussianObject (Yang et al., 2024) creatively draws ideas from ControlNet (Zhang et al., 2023) and allows for refinement of the generated Gaussian splatting-based rendering. Furthermore, to generate the 3D object from sparse views, the method relies on the visual hull instead of a conventional SfM pipeline. ESC3D (Wu et al., 2025) proposes a novel method for controllable text-to-3D generation. Their pipeline allows for richer prompt generation and also allows for additional priors for conditioning the generation. Hallo3D (Wang et al., 2024)p focuses on hallucination alleviation. The authors propose the usage of a generation-detection-correction paradigm and leverage modern large multi-modal models for guiding the generation.

In parallel, recent advancements (Hong et al., 2023; Xiang et al., 2024) have explored large-scale training on comprehensive 3D datasets, such as Objaverse (Deitke et al., 2023), to improve text-to-3D and image-to-3D models. While these feedforward methods achieve impressive speed and consistency, SDS-based pipelines remain attractive due to their ability to flexibly adapt to unseen prompts and leverage the rich priors of pre-trained diffusion models. However, SDS-based methods continue to face challenges in training efficiency, 3D consistency, and reconstruction quality. To address these limitations, we propose MVGaussian, which improves efficiency, quality, and robustness for text-to-3D generation using 3D Gaussian Splatting.

## 3 Background

### 3.1 Diffusion process

Diffusion has emerged as a pivotal approach in generative modeling, particularly for text-to-image generation tasks (Ramesh et al., 2022; Zhang et al., 2023; Saharia et al., 2022). Recent advancements show that diffusion

models not only surpass traditional generative adversarial networks (GANs) (Goodfellow et al., 2014) in image quality but also provide improved training stability and convergence (Dhariwal & Nichol, 2021; Ho et al., 2020a). These models simulate the reverse process of diffusion, starting with corrupted input data and progressively reconstructing it back to the original form. In text-to-image applications, the diffusion process is typically applied in the latent space, which reduces dimensionality, accelerates computations, and lowers memory requirements while preserving essential data features for high-quality generation.

## 3.2 Score Distillation Sampling

Score Distillation Sampling (SDS) (Poole et al., 2022) is an optimization technique that integrates pre-trained 2D diffusion models into the synthesis of 3D objects. Instead of training a 3D generative model directly, SDS optimizes a 3D representation so that its 2D projections, when rendered from different viewpoints, match the image distribution learned by the diffusion model. This ensures that the generated 3D object aligns with the expected visual features of the target category or text prompt.

A key component of SDS is the score function, defined as $s_\phi(\mathbf{z}_t; \theta) = -\hat{\epsilon}_\phi(\mathbf{z}_t; \theta)/\sigma_t$. Here, $\mathbf{z}_t$ represents the latent variable at time step $t$, which is a noisy version of an image. The term $\epsilon_\phi(\mathbf{z}_t; \theta)$ is the noise prediction function, which estimates the noise added during the diffusion process, while $\sigma_t$ is the noise level at step $t$, controlling the variance of noise introduced in the forward diffusion process. These score functions represent gradients of the log probability density, guiding the optimization towards regions of higher likelihood under the diffusion model's learned distribution.

The gradient of the SDS loss function $\mathcal{L}_{\text{SDS}}$ is computed as

$$\nabla_\theta \mathcal{L}_{\text{SDS}} = \mathbb{E}_{t,\epsilon} \left[ w(t)(\hat{\epsilon}_\phi(\mathbf{z}_t; \theta) - \epsilon)\frac{\partial \mathbf{x}}{\partial \theta} \right], \tag{1}$$

where $w(t)$ is a weighting function that depends on the time step $t$, $\epsilon$ is the actual noise added in the forward process, and $\mathbf{x}$ is the rendered 2D image of the 3D model, parameterized by $\theta$. This formulation enables the optimization to refine the 3D model such that its rendered 2D views match the statistical properties of realistic images, ensuring high-quality 3D synthesis guided by the diffusion model.

## 3.3 3D Gaussian Splatting

The seminal work presented by Kerbl et al. (2023) introduces an explicit approach for representing and rendering three-dimensional objects using Gaussian functions as the fundamental building blocks. 3D Gaussian splatting employs continuous Gaussian distributions to define the geometry and appearance of a 3D model. Each Gaussian $G$ is characterized by the position of its center or mean $\mu \in \mathbb{R}^3$, color $c \in \mathbb{R}^3$, opacity $\sigma \in [0,1]$, and a full covariance matrix $\Sigma \in \mathbb{R}^{3 \times 3}$. This covariance matrix $\Sigma$ is decomposed into a rotation matrix $R$ and a scaling matrix $S$ as

$$\Sigma = RSS^T R^T \tag{2}$$

to ensure valid optimization, as directly optimizing $\Sigma$ via gradient descent can produce non-positive semi-definite matrices. The parameters $R$ and $S$ are stored and optimized independently. The matrix $S$ consists of scale values $s_1, s_2, s_3$, along the different axes $x, y, z$ of the Gaussian and the minimum scale is denoted as $s_g$. Consequently, a 3D Gaussian can be defined as

$$G(x) = \exp\left(-\frac{1}{2}(x-\mu)^T \Sigma^{-1}(x-\mu)\right), \tag{3}$$

and the influence of a Gaussian splat is formulated as $\alpha(x) = \sigma G(x)$, where $x \in \mathbb{R}^3$ is a point in 3D space.

To render a scene using Gaussian splats, the 3D Gaussians are projected onto a 2D image plane. The contribution of each splat to the final image is determined by integrating the Gaussian over the pixels it influences. The final color of a rendered pixel is a combination of the influences from all ordered point

samples along a ray that project to the pixel

$$C = \sum_i c_i \alpha_i \prod_{j=1}^{i-1} (1 - \alpha_j).$$

(4)

Unlike implicit representations used in NeRF models, Gaussian splatting is an explicit method that requires a mechanism to manage the number of Gaussians. This is achieved through a unique densification and pruning scheme, discussed subsequently.

## 4 Method

Our proposed method leverages MVDream along with a novel densification and pruning scheme to reduce the Janus problem. The densification and pruning scheme utilizes multi-view guidance and leverages back-projected points from the estimated depth on the fly to optimize the Gaussians. We observe that surface alignment techniques, as demonstrated in SuGaR (Guédon & Lepetit, 2024), are beneficial for improving surface fidelity and mesh extraction. In contrast, our approach is designed for SDS-based text-to-3D generation, where we introduce a **regularization term** that allows for flattening the Gaussians during the learning process itself. This flattening facilitates more effective pruning and improves geometry consistency during optimization. We primarily rely on multi-view guidance to mitigate the Janus issue and ensure consistent 3D reconstruction across different viewpoints. Additionally, we refine the densification strategy and enforce surface proximity for the generated Gaussians.

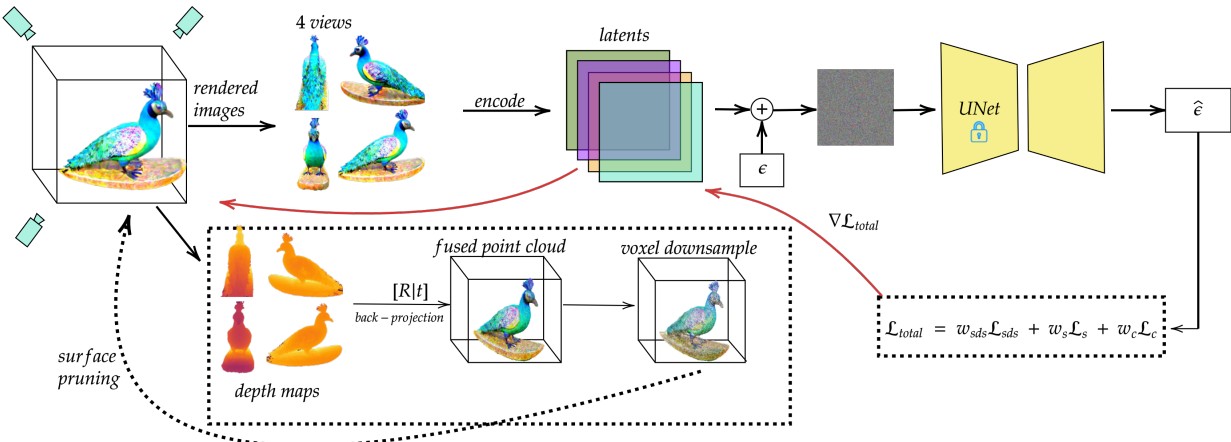

Figure 1: **Overview of our MVGaussian framework:** Our approach begins with the random initialization of Gaussians within a unit sphere, refined iteratively using an SDS-based optimization strategy. Gaussians are optimized near the true surface, moving toward the pseudo surface while pruning those farther away. Each iteration renders four views with random azimuth angles encoded into the latent space. Gaussian noise is added and subsequently denoised by a UNet to compute the losses. The resulting gradients update the Gaussians, forming a feedback loop that integrates fused point cloud data and voxel downsampling to improve accuracy.

The overall framework, as depicted in Figure 1, showcases how our approach integrates these components to produce more consistent and unified 3D reconstructions. This method not only addresses the shortcomings of prior techniques but also enhances the efficiency and quality of the generated 3D models.

### 4.1 Multi-view guidance for consistent 3D generation

SDS-based approaches for text-to-3D generation often suffer from multi-face or the Janus problem (Poole et al., 2022). This issue arises as the diffusion models are trained on 2D images and lack a true understanding

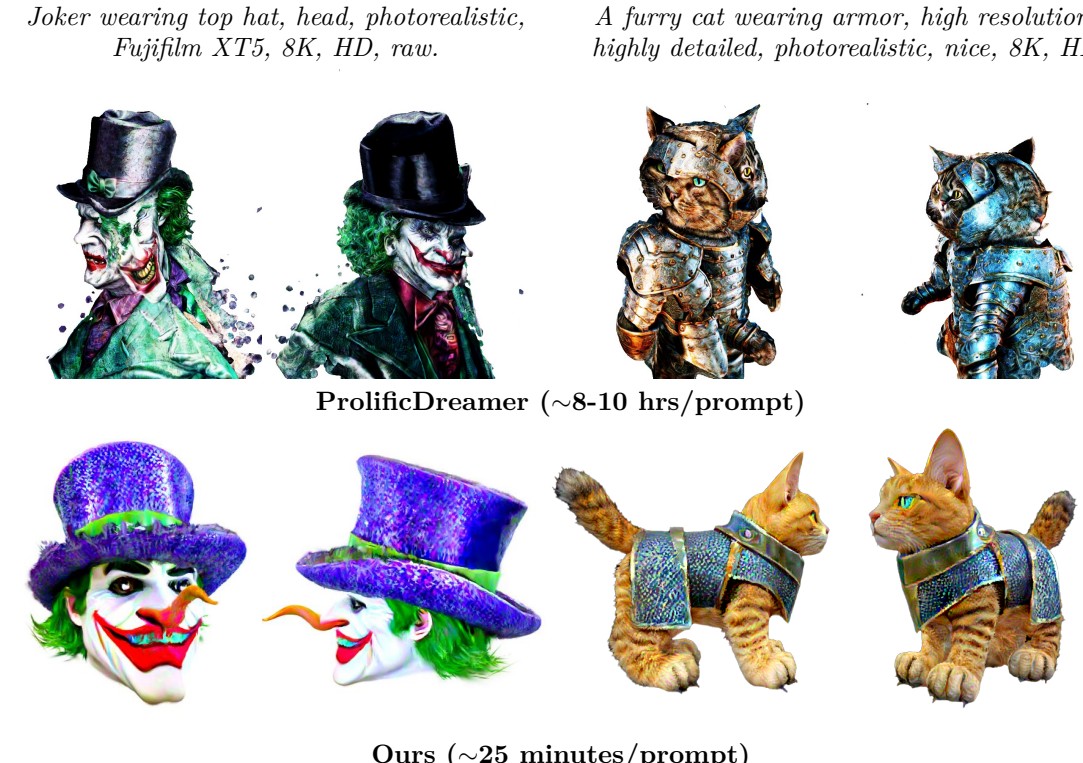

*Joker wearing top hat, head, photorealistic, Fujifilm XT5, 8K, HD, raw.*  *A furry cat wearing armor, high resolution, highly detailed, photorealistic, nice, 8K, HD*

**ProlificDreamer (∼8-10 hrs/prompt)**

**Ours (∼25 minutes/prompt)**

Figure 2: Comparison of ProlificDreamer (top, 8–10 hrs/prompt) and our 3DGS-based method (bottom, ∼25 mins/prompt). Our method produces more coherent, photorealistic results with fewer artifacts.

of the 3D world. Consequently, while rendered images might appear plausible from different viewpoints, they often fail to represent a consistent and unified 3D object. Several strategies have been developed to address the Janus problem. Notably, Zero123 (Liu et al., 2023) and MVDream (Shi et al., 2024) have made significant strides by fine-tuning pre-trained diffusion models on 3D data. Zero123 predicts multi-view images conditioned on a reference image and camera position, while MVDream fine-tunes diffusion models to generate multi-view images from text inputs. Despite these advancements, these methods do not completely resolve the Janus problem, as the generated multi-view images often lack the exact consistency needed for unified 3D models since they lack precise symmetry and high-level detail correspondence across the generated views. However, they do provide reliable guidance for SDS-based approaches.

To address the Janus problem, we integrate the strengths of MVDream as the primary guidance mechanism within our framework. By adopting MVDream, we leverage its ability to generate multi-view images from textual inputs, thereby providing robust guidance for our 3D models. As shown in Figure 1, we render four views around the current object at each training step. We then map these multi-view images to the latent space and perform the noising and denoising steps. Similar to Dreamfusion, we adopt classifier-free guidance (CFG) proposed by Ho & Salimans (2021) to enhance the quality of generated 3D models. CFG adjusts the score function to favor regions with a higher ratio of conditional to unconditional density, using a guidance scale parameter $\omega$.

### 4.2 Gaussian Alignment for Optimal Geometry

We propose a novel regularization term that is computationally efficient and facilitates on-the-fly optimization of Gaussians. Guédon & Lepetit (2024) have explored aligning Gaussians to the surface by minimizing a Signed Distance Function (SDF)-based regularization term $\mathcal{R}$, which requires precomputing the SDF before appearance modeling, as shown in Fantasia3D (Chen et al., 2023b). However, such methods introduce additional computational overhead and are not well-suited for score distillation strategies. Additionally, existing

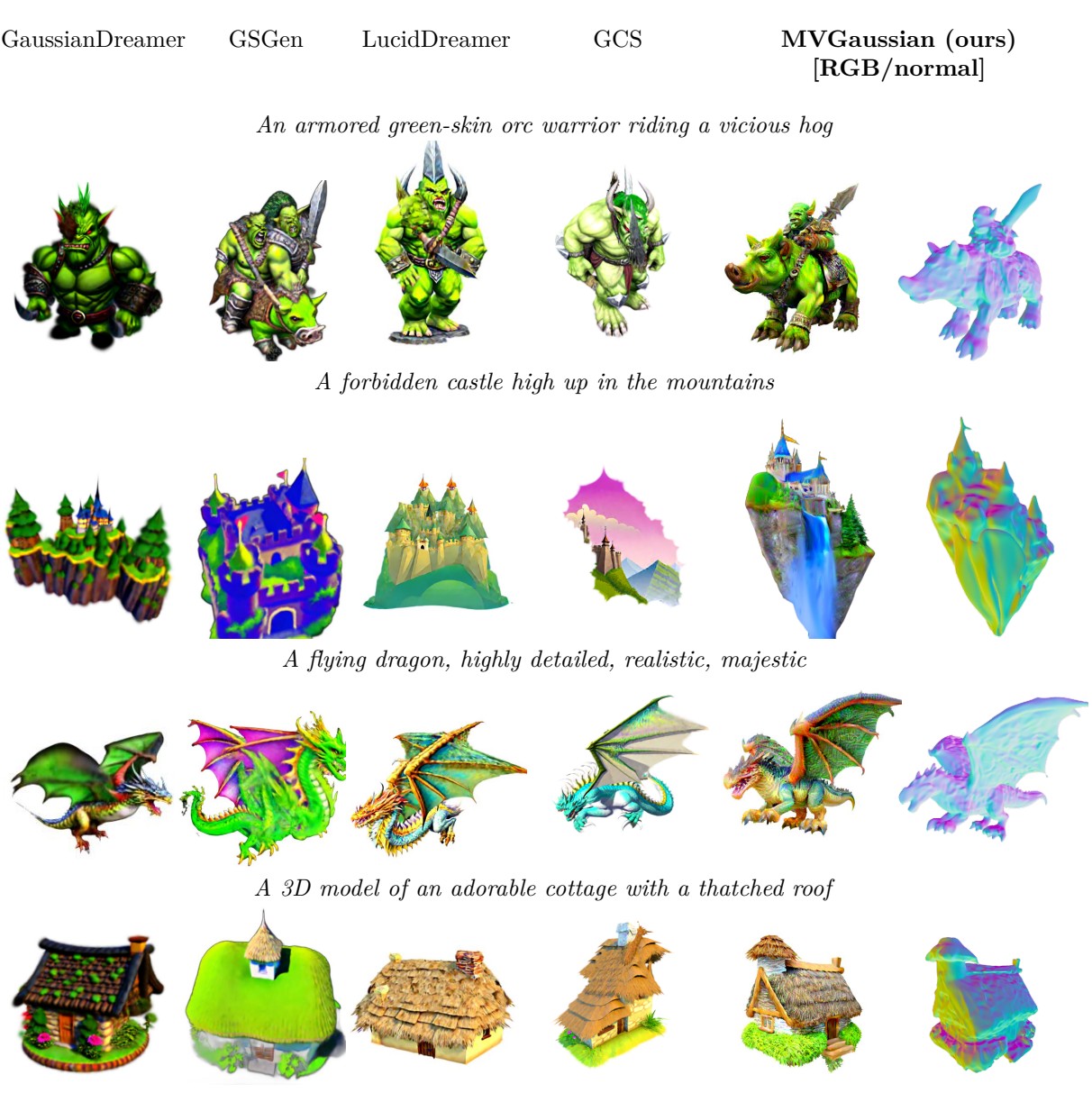

Figure 3: Extensive qualitative comparisons on object and scene prompts. We observe consistent improvement across prompts and baselines, demonstrating the effectiveness of our densification approach.

GaussianDreamer     GSGen     LucidDreamer     GCS     **MVGaussian (ours)**
**[RGB/normal]**

*A blue jay sitting on a willow basket of macarons*

*Medieval soldier with shield and sword, fantasy, game,*
*character, highly detailed, photorealistic, 4K, HD*

*Jack Sparrow wearing sunglasses, head, photorealistic, 8K, HD, raw*

*A peacock standing on a surfing board, highly detailed, majestic*

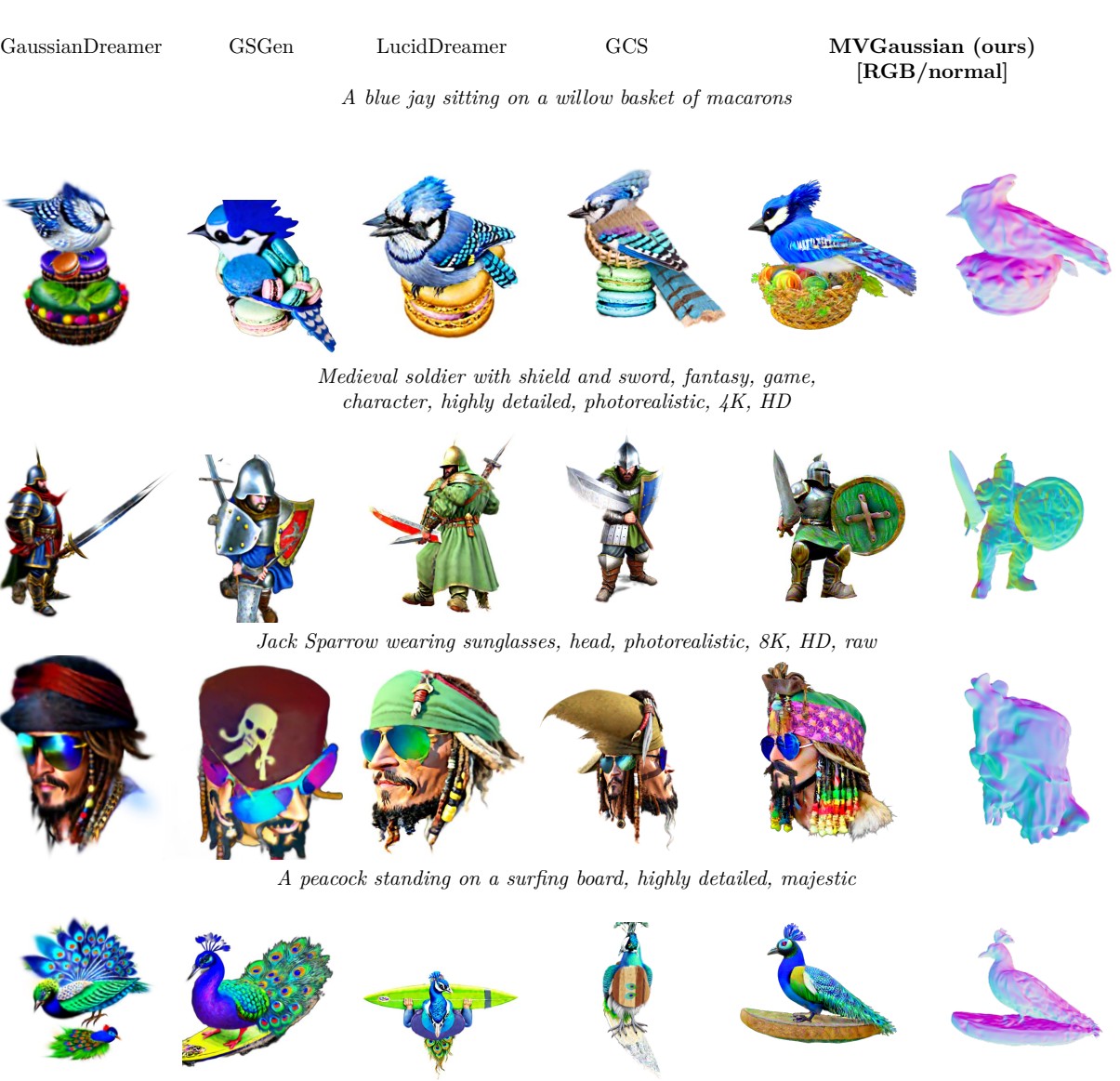

Figure 4: Additional qualitative comparisons on character and object prompts. Our method achieves visually coherent, high-fidelity 3D results and outperforms all the baselines.

Ours | Fantasia3D | Prolific Dreamer | LucidDreamer | Magic3D
($\sim$ 25 mins) | ($\sim$ 1 hr) | ($\sim$ 8 hrs) | ($\sim$ 35 mins) | ($\sim$ 1 hr)

*A DSLR photo of the Imperial State Crown of England.*

*A DSLR photo of a Schnauzer wearing a pirate hat.*

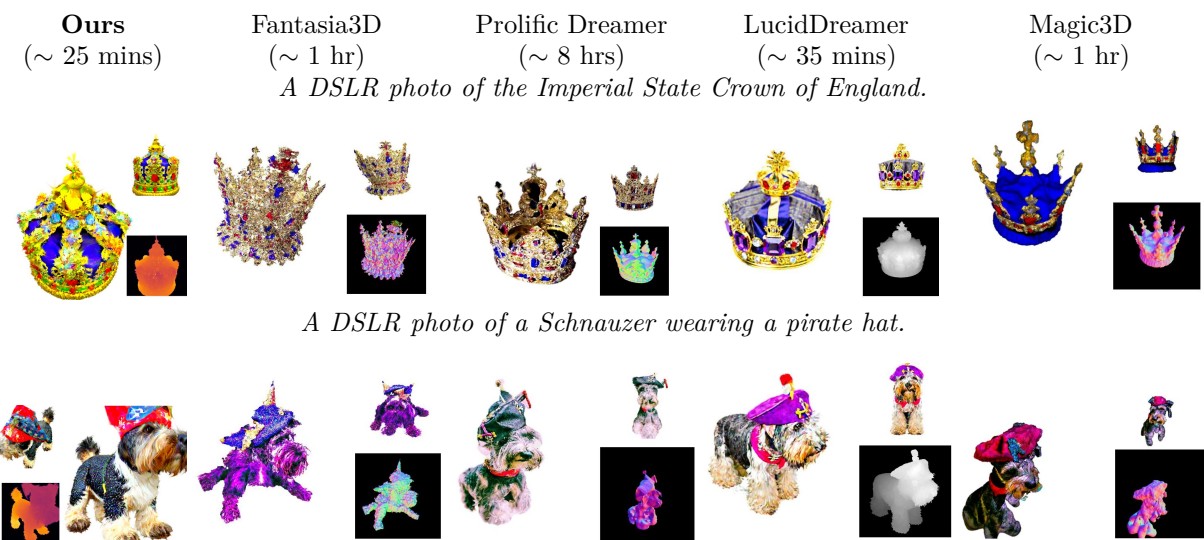

Figure 5: Additional qualitative comparisons with several state-of-the-art methods.

approaches often apply regularization as a post-processing step rather than integrating it into the optimization process. In contrast, our method directly optimizes Gaussian alignment during training, improving efficiency and adaptability.

Suppose we have a true surface of the 3D scene described by a given text prompt; we want the Gaussians to lie on the surface to capture fine details and intricate geometries, resulting in high-fidelity reconstructions. For any point $x \in \mathbb{R}^3$ on the surface we can find the Gaussian $g^*$ that has the most significant influence on the appearance of $x$:

$$g^* = \arg \max_g \left[ \sigma_g \exp \left( -\frac{1}{2}(x - \mu_g)^T \Sigma_g^{-1} (x - \mu_g) \right) \right]. \tag{5}$$

Ideally, we want the center of $g^*$ to be close to the surface point $x$, i.e., $\mu_g^* \approx x$, so that the exponent approaches zero:

$$T = (x - \mu_{g^*})^T \Sigma_{g^*}^{-1} (x - \mu_{g^*}) \to 0. \tag{6}$$

Minimizing this exponent term encourages the Gaussians to be close to the surface. When a Gaussian lies on the surface, it should be flattened to accurately represent the geometry of the 3D object. Therefore, one of the three scales of the Gaussian $g^*$ should be close to 0. We can express $\Sigma_{g^*}$ in terms of its eigenvalues and eigenvectors as

$$\Sigma_{g^*} = U \Lambda U^T,$$

where $U$ is the matrix of eigenvectors and $\Lambda$ is the diagonal matrix of eigenvalues.

$$U = [\mathbf{v}_1 \, \mathbf{v}_2 \, \mathbf{v}_3]; \quad \Lambda = \begin{pmatrix} \lambda_1 & 0 & 0 \\ 0 & \lambda_2 & 0 \\ 0 & 0 & \lambda_3 \end{pmatrix}. \tag{7}$$

The inverse of the covariance matrix $\Sigma_{g^*}$ is given by

$$\Sigma_{g^*}^{-1} = U \Lambda^{-1} U^T = \begin{pmatrix} \mathbf{v}_1 & \mathbf{v}_2 & \mathbf{v}_3 \end{pmatrix} \begin{pmatrix} \frac{1}{\lambda_1} & 0 & 0 \\ 0 & \frac{1}{\lambda_2} & 0 \\ 0 & 0 & \frac{1}{\lambda_3} \end{pmatrix} \begin{pmatrix} \mathbf{v}_1^T \\ \mathbf{v}_2^T \\ \mathbf{v}_3^T \end{pmatrix}. \tag{8}$$

Substituting into the exponent term $T$ in Eq. (6), we get

$$T = \sum_i \frac{1}{\lambda_i}(x - \mu_g)^T \mathbf{v}_i \mathbf{v}_i^T (x - \mu_g), \tag{9}$$

where $\mathbf{v}_i$ are the eigenvectors in $U$.

Let $j \in \{1, 2, 3\}$ be the direction of the smallest scale of a Gaussian, whose corresponding eigenvalue $\lambda_j$ and the scale $s_j$ is also minimal. Consequently, when the Gaussian is nearly flat (i.e., $s_j$ is small thus $\lambda_j$ is small), the exponent is dominated by this direction and can be approximated by the term

$$T \approx \frac{1}{\lambda_j}(x - \mu_g)^T \mathbf{v}_j \mathbf{v}_j^T (x - \mu_g). \tag{10}$$

Additionally, to ensure that Gaussians are flattened when they align with the surface, we directly penalize the smallest scale $s_g$ of each Gaussian. This loss term enforces that at least one of the scales collapses towards zero when a Gaussian lies on the surface. Thus, the final loss function for flattening Gaussians while ensuring they stay close to the surface is given by:

$$\mathcal{L}_s = \sum_g \left[ \frac{1}{\lambda_g}(x - \mu_g)^T \mathbf{v}_g \mathbf{v}_g^T (x - \mu_g) + |s_g| \right]. \tag{11}$$

where $\lambda_g$, $\mathbf{v}_g$ are the eigenvalue, eigenvector corresponding to the smallest scale $s_g$ of the Gaussian $g$, $x$ is sampled from on the pseudo surface constructed from the rendered depth maps.

To enforce smoothness in both the geometry (depth) and appearance (color), we introduce the **smoothness loss**, which penalizes abrupt changes in depth and color while preserving object boundaries. The loss leverages image gradients to weight the smoothness penalty adaptively, ensuring strong regularization in homogeneous regions and reduced regularization near edges.

Given depth maps $D \in \mathbb{R}^{B \times H \times W}$ and corresponding RGB images $I \in \mathbb{R}^{B \times 3 \times H \times W}$, the smoothness loss is defined as

$$\mathcal{L}_c = \sum_{i=1}^{B} \sum_{u,v} \left( w_u(u, v) \cdot \|\nabla_u D(u, v)\| + w_v(u, v) \cdot \|\nabla_v D(u, v)\| \right), \tag{12}$$

with

$$w_v(u, v) = \exp\left(-\|\nabla_u I(u, v)\|\right), \quad w_v(u, v) = \exp\left(-\|\nabla_v I(u, v)\|\right), \tag{13}$$

where $\nabla_u I(u, v)$ and $\nabla_v I(u, v)$ are the horizontal and vertical gradients of the RGB image, averaged across the color channels:

$$\nabla_u I(u, v) = \frac{1}{3} \sum_{c=1}^{3} |I_c(u + 1, v) - I_c(u, v)|, \quad \nabla_v I(u, v) = \frac{1}{3} \sum_{c=1}^{3} |I_c(u, v + 1) - I_c(u, v)|. \tag{14}$$

The depth gradients $\nabla_u D(u, v)$ and $\nabla_v D(u, v)$ measure the horizontal and vertical changes in depth values, respectively.

The smoothness loss couples color and geometry by using image gradients as proxies for scene boundaries. By weighting depth gradients inversely proportional to image gradients, the model enforces smoothness in regions with uniform color while preserving sharp transitions near edges. This adaptive weighting reduces the penalty on depth variations in high-gradient areas, ensuring that fine details in both geometry and appearance are retained.

The final loss is a weighted sum of the individual losses, balancing their contributions. The overall objective is defined as

$$\mathcal{L}_{total} = w_{sds}\mathcal{L}_{sds} + w_s\mathcal{L}_s + w_c\mathcal{L}_c. \tag{15}$$

In our experiments, we set the weighting parameters to prioritize the contributions of the smoothness and regularization terms relative to the SDS loss. Specifically, we use $w_{\text{sds}} = 1$, as the magnitude of the SDS loss is significantly higher than the other terms, ensuring its influence remains balanced without overshadowing other contributions. The weights for the smoothness loss and regularization loss are set to $w_{\text{s}} = w_{\text{c}} = 200$ to enforce strong geometric and appearance constraints that enhance surface fidelity and color consistency. These settings were chosen empirically to achieve high-quality reconstructions while maintaining efficient convergence.

## 4.3 Surface densification and pruning

In this section, we relook at the densification approach used in 3DGS and discuss our strategy to overcome the limitations of the existing methods. Naive 3D Gaussian splatting methods densify the Gaussians based on the gradient of the Gaussian centers and the scales of the Gaussians. While this approach is straightforward, it presents several significant drawbacks. One of the primary challenges lies in defining an appropriate threshold value for the gradient. If the threshold value is set too high, fewer Gaussians are added to the scene, leading to a lack of detail in the reconstructed model. Conversely, if the threshold value is set too low, the number of Gaussians increases significantly. This not only hinders the learning speed but also impedes the convergence of the model due to the excessive computational load.

We propose an intuitive method that utilizes the rendered image and depth to backproject the rendered pixels to the world using camera parameters. This allows us to progressively reconstruct the surface of the 3D model. We densify the Gaussians that are close to the surface, allowing the model to gradually reconstruct the missing parts and speed up the training time due to the significantly reduced number of Gaussians to update. Mathematically, we define the backprojection of a pixel $p$ with depth $d$ and camera parameters $K$ (intrinsic matrix) and $[R|t]$ (extrinsic matrix) as follows:

$$P = R^{-1}(K^{-1}p'd - t) \tag{16}$$

where $p'$ is the homogeneous coordinate of $p$. Let $\{P_i\}$ be the set of all backprojected points. We then define the distance $D_g$ of a Gaussian $g$ from the surface as the Euclidean distance between the Gaussian center $\mu_g$ and the closest backprojected point:

$$D_g = \min_{P_i} \|\mu_g - P_i\| \tag{17}$$

We prune Gaussians for which $D_g$ exceeds a threshold $\epsilon = 0.02$. This approach allows us to significantly reduce the number of Gaussians and improve the efficiency and quality of the final 3D reconstruction.

| Prompt | Human evaluation scores | | | | No. Gaussians | |
| --- | --- | --- | --- | --- | --- | --- |
| | **GaussianDreamer** | **GSGen** | **LucidDreamer** | **Ours** | **Naive** | **Ours** |
| *A blue jay sitting on a willow basket of macarons* | 3.23 | 2.89 | 3.07 | **4.65** | 16.2 | **1.1** |
| *A flying dragon, highly detailed, realistic, majestic* | 2.51 | 2.12 | 3.45 | **4.81** | 24.5 | **1.2** |
| *An armored green-skin orc warrior riding a vicious hog* | 3.12 | 2.94 | 3.21 | **4.56** | 22.7 | **1.3** |
| *A forbidden castle high up in the mountains* | 3.24 | 2.46 | 2.87 | **4.45** | 24.2 | **1.2** |
| *A peacock standing on a surfing board, highly detailed, majestic* | 2.02 | 3.56 | 2.95 | **4.64** | 18.8 | **1.1** |
| *Jack Sparrow wearing sunglasses, head, photorealistic, 8k, HD, raw* | 4.01 | 3.21 | 4.15 | **4.45** | 16.7 | **0.9** |
| *Medieval soldier with shield and sword, fantasy, game, character, highly detailed, photorealistic, 4K, HD* | 3.5 | 2.57 | 3.64 | **4.32** | 17.3 | **1.2** |
| *A 3D model of an adorable cottage with a thatched roof* | 3.45 | 1.89 | 3.42 | **3.89** | 16.9 | **1.2** |

Table 1: Comparison of different methods based on the provided prompts. The table includes human evaluation scores for each method, along with the number of Gaussians (in millions) utilized by both the naive and our approach. The highest scores in each row are highlighted in bold.

## 5 Experiments and Results

We generate a variety of outputs using a diverse set of prompts and observe that our method significantly outperforms other 3DGS-based approaches within the same time constraints. Our approach not only achieves a higher level of detail but also exhibits fewer artifacts compared to existing methods.

In this work, we focus on SDS-based approaches that utilize the 3DGS representation. Existing NeRF-based approaches have been thoroughly studied and compared in previous work Yi et al. (2024); Chen et al. (2024); Liang et al. (2023), demonstrating that they suffer from slow convergence and inferior quality compared to recent Gaussian splatting-based methods. Figure 2 illustrates that our method outperforms ProlificDreamer, one of the state-of-the-art NeRF-based approaches, in terms of visual quality, geometric accuracy, and computational efficiency.

We primarily compare our method against several state-of-the-art 3DGS-based techniques, including GaussianDreamer Yi et al. (2024), GSGen (Chen et al., 2024), and LucidDreamer (Liang et al., 2023). As shown in Figures 3 and 4, our method produces brighter colors and sharper structures, achieving a photorealistic appearance. Our findings indicate that methods such as GSGen and LucidDreamer, which rely heavily on Point-e (Nichol et al., 2022) initialization, struggle to produce high-quality results if the initial point cloud is suboptimal. For instance, GSGen still exhibits the Janus problem, particularly evident in the **Jack Sparrow** model, while LucidDreamer produces extraneous arms holding the surfboard in the **Peacock** model. Even in more recent methods such as GCS (Li et al., 2024), we continue to observe Janus artifacts (e.g., in the **Jack Sparrow** example), along with weaker prompt alignment, such as the missing basket in the **Blue Jay** prompt.

In contrast, our method excels at generating photorealistic results with a higher level of detail compared to other approaches and better prompt adherence. We observe that Point-e initialization often leads to missing structures, such as the absence of the hog in the **Green Orc** model and the lack of mountains in the **Castle** model generated by GSGen. As shown in Figures 3 and 4, our method consistently uses approximately 1M Gaussians to render higher-quality details, while other methods require roughly $15-20$ times more Gaussians to achieve similar results, as shown in Figure 5. Furthermore, in most cases, our method achieves a higher CLIP score.

Our training process involves $10,000$ steps, with the densification process starting at $1,000$ steps and occurring every 200 iterations on an NVIDIA A100 GPU. The code is written in PyTorch, and the entire process takes approximately 25 minutes per prompt.

## 6 User Study

To further evaluate the performance of our method, we conducted a user survey with 42 participants, as shown in Figure 6. Each participant was asked to rate eight outputs generated by different text-to-3D models on a scale from 1 to 5 (higher is better). As shown in Table 1, the average human evaluation scores demonstrate that our method significantly outperforms other methods by a wide margin. Additionally, participants were asked to assess the models based on visual quality, geometry, and prompt alignment of the generated content. As illustrated in Figure 6, our method achieved superior scores across all metrics, with average ratings of 4.45 for visual quality, 4.65 for geometry, and 4.78 for prompt alignment. In contrast, the next best-performing method, LucidDreamer, received average scores of 3.11, 2.88, and 2.45, respectively. These human evaluation results underscore our method's ability to produce more accurate and aesthetically pleasing 3D models, highlighting its effectiveness in overcoming the limitations of existing approaches. Further, we demonstrate the effectiveness of our proposed method via ablations in the Appendix in Figure 12.

## 7 Limitations

While our method generates high-quality results in a relatively short time, it has several limitations. The failure cases shown in Figure 7 highlight issues specific to our approach, particularly the occurrence of spiky Gaussian artifacts. These artifacts, noticeable in models such as the **Plate of cookies** and the **Iron Man**,

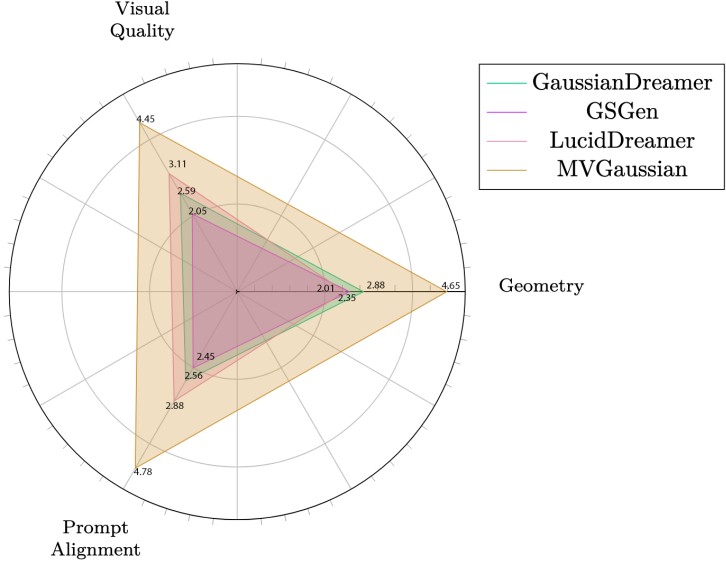

Figure 6: Evaluation of various aspects of the generated 3D content across different text-to-3D models based on human assessments.

may result from the flatness regularization used in our method. This regularization could inadvertently introduce sharp spikes on the model surfaces, disrupting their smoothness.

Additionally, the ***Warrior on a horse*** and the ***Plate of cookies*** exhibit unrealistic textures that detract from their fidelity. For instance, in the ***Warrior on a horse***, the horse's body is covered with unnatural floral or abstract patterns that do not resemble realistic fur or skin. While visually interesting, these patterns disrupt the semantic alignment with the prompt, making the output appear more like a surreal painting than a 3D render. Similarly, in the ***Plate of cookies***, the plate is rendered with an irregular, multicolored texture instead of the expected smooth white ceramic appearance. This inconsistency may stem from an over-reliance on guidance that prioritizes artistic styles over photorealism.

These issues could be mitigated with longer training, allowing the model to better converge and reduce such artifacts. Despite these occasional imperfections, our method performs better across a wider range of prompts compared to other approaches, demonstrating its robustness and effectiveness in generating high-quality 3D content.

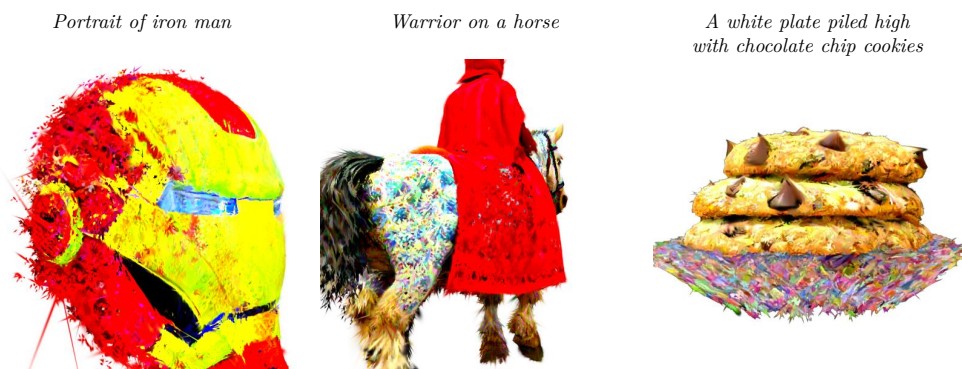

Figure 7: Failure cases, usually contain spike-like artifacts or irregular texture.

# 8 Conclusion

We present an intuitive and elegant method for high-quality text-to-3D renderings using depth maps without external supervision. Our approach employs the back-projection of screen space points to 3D for filtering Gaussians and leverages multi-view diffusion guidance along with surface alignment to achieve superior results. This technique not only produces higher-quality renderings in significantly less time but also demonstrates robustness across diverse text prompts. Our method generates highly detailed renderings using Gaussian splatting in under half an hour, striking an optimal balance between quality and speed, unlike other methods. This establishes a rapid, SDS-based high-quality rendering scheme.

# 9 Ethic concern and societal impact

Generative models for 3D content synthesis raise a number of ethical considerations, which apply to our method as well. Because MVGaussian uses MVDream (Shi et al., 2024) as a multi-view diffusion prior, it inherits any biases or limitations present in MVDream and its underlying models, such as Stable Diffusion (Rombach et al., 2022b). As these diffusion models are trained on large-scale web data, they may reflect undesirable biases or produce outputs that are inappropriate or unlicensed. It is important to carefully consider the text prompts and model outputs, and to implement appropriate content moderation to avoid generating harmful or unethical media. We encourage responsible use of this method and adherence to ethical standards in its deployment.

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

# A  Appendix

## A.1  Quantitative Evaluation

| Prompt | CLIP score | | | | |
|---|---|---|---|---|---|
| | GaussianDreamer | GSGen | LucidDreamer | GCS | Ours |
| *A blue jay sitting on a willow basket of macarons* | 0.28 | 0.27 | 0.33 | 0.32 | **0.34** |
| *A flying dragon, highly detailed, realistic, majestic* | 0.30 | 0.30 | 0.30 | 0.28 | **0.32** |
| *An armored green-skin orc warrior riding a vicious hog* | 0.29 | **0.31** | 0.29 | 0.29 | **0.31** |
| *A forbidden castle high up in the mountains* | **0.32** | 0.25 | 0.27 | 0.25 | 0.30 |
| *A peacock standing on a surfing board, highly detailed, majestic* | 0.31 | 0.29 | 0.29 | 0.30 | **0.33** |
| *Jack Sparrow wearing sunglasses, head, photorealistic, 8k, HD, raw* | 0.27 | **0.33** | 0.29 | 0.29 | 0.31 |
| *Medieval soldier with shield and sword, fantasy, game, character, highly detailed, photorealistic, 4K, HD* | 0.25 | 0.29 | 0.26 | 0.28 | **0.31** |
| *A 3D model of an adorable cottage with a thatched roof* | 0.32 | 0.31 | **0.34** | **0.34** | 0.31 |

Table 2: Comparison of different methods based on the given prompt. The CLIP score is computed for 15 views generated from the 3D model of each method, then averaged. We also compute the number of Gaussians for the naive method, which does not use the surface densification proposed by our method.

In Table 2, we compare the performance of our method against GaussianDreamer (Yi et al., 2024), GSGEN (Chen et al., 2024), LucidDreamer (Liang et al., 2023) and GCS Li et al. (2024) using the CLIP score, averaged over 15 views generated from 3D models for each prompt shown in Figures 3 and 4. Our method consistently achieves the highest or near-highest scores across various prompts, such as **Blue jay**, **Peacock**, and **Dragon**, indicating superior text-image alignment. However, despite the superior visual quality, the CLIP scores do not show a significant improvement for all prompts, and in some cases (**Castle**, **Jack Sparrow**, **Cottage**), our CLIP scores are even lower. This discrepancy arises because the CLIP score, while useful for measuring 2D image-text alignment, is not a reliable metric for evaluating the performance of text-to-3D models. The CLIP score does not fully capture the fidelity, coherence, or geometric accuracy of the 3D models across different views, leading to potential underestimation of the quality improvements introduced by our method, e.g., the Jacksparrow model, despite having Janus problem in GSGen scores higher than ours. Therefore, additional metrics beyond the CLIP score, such as human evaluation in the form of user studies as we conducted, are necessary to thoroughly assess the overall quality of text-to-3D model generation.

## A.2  Ablation studies

**Ablation on $\mathcal{L}_s$.** The ablation study presented in Figure 8 examines the impact of the surface regularization loss $\mathcal{L}_s$ on the quality and consistency of 3D reconstructions. The top row displays results generated without $\mathcal{L}_s$, while the bottom row shows outputs with $\mathcal{L}_s$ applied. The comparison highlights that models utilizing the $\mathcal{L}_s$ loss produce more refined and visually compelling results characterized by clearer details and reduced artifacts. For instance, in the **Wine glass** example, the model with regularization achieves smoother surfaces and more consistent transparency effects. Likewise, the **Crown** with regularization displays a more polished and realistic appearance, with additional details and enhanced structural elements. The **Cottage** and **Jack Sparrow** examples also benefit from regularization, showing sharper details and more accurate textures, leading to a more realistic and appealing visual representation.

Figure 9 provides a qualitative comparison of **surface reconstruction** quality with and without the $\mathcal{L}_s$ regularization loss. The addition of $\mathcal{L}_s$ results in smoother and more consistent surfaces, as seen in the normal maps on the right side of each example. In the **Cottage** example, the surface reconstructed without $\mathcal{L}_s$ exhibits noisy and uneven geometry, whereas the version with $\mathcal{L}_s$ yields a cleaner, well-aligned surface. Similarly, in the **Jack Sparrow** example, the hat and facial details show noticeable geometric noise and irregularities without $\mathcal{L}_s$, which are substantially reduced when the regularization is applied.

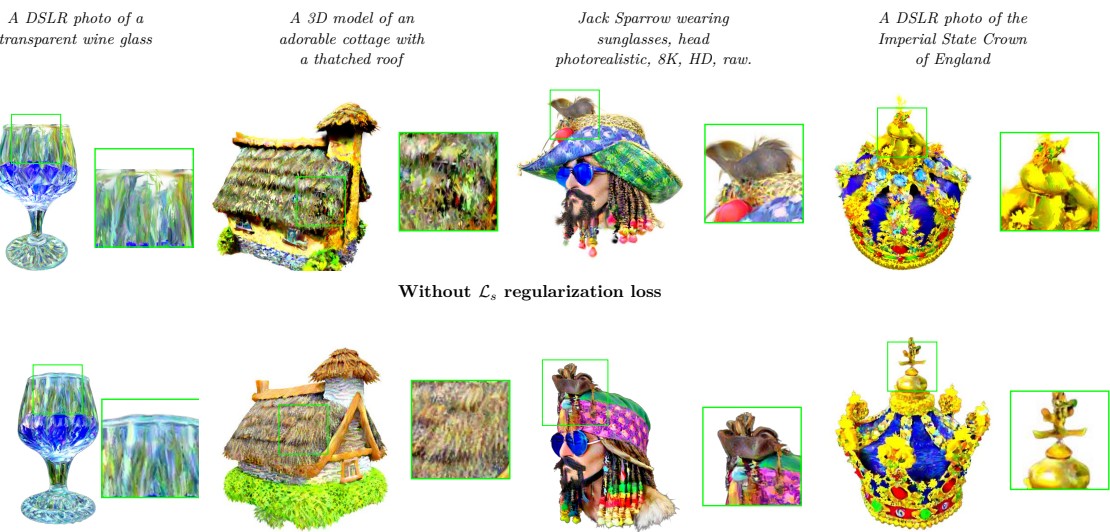

*A DSLR photo of a transparent wine glass*  ·  *A 3D model of an adorable cottage with a thatched roof*  ·  *Jack Sparrow wearing sunglasses, head photorealistic, 8K, HD, raw.*  ·  *A DSLR photo of the Imperial State Crown of England*

**Without $\mathcal{L}_s$ regularization loss**

**With $\mathcal{L}_s$ regularization loss**

Figure 8: Ablation study on the surface and flattening loss. We observe smoother, better textured surfaces with the $\mathcal{L}_s$ term in the loss.

**Ablation on $s_g$ term.** To further analyze the contribution of the $s_g$ term of $\mathcal{L}_s$, we conduct an ablation study as shown in Figure 10. This experiment isolates the effect of explicitly encouraging the flattening of Gaussians along the surface normal direction. Without the $s_g$ term, the reconstructed shapes tend to exhibit thicker, more blob-like geometry, which blurs or oversmooths fine structures. In contrast, applying the $s_g$ term results in sharper representations of thin and delicate features. For example, in the ***Fluffy dog*** example, the fur is better captured with clear texture and volume when the flattening term is used, making it more aligned with the *fluffy* aspect of the prompt. The ***Peacock*** tail also exhibits more refined feather structures, rather than a clumped appearance. Similarly, in the ***Quill and ink*** example, the quill becomes thinner and more naturally shaped with the flattening regularization.

**Ablation on $\mathcal{L}_c$.** Figure 11 presents an ablation study analyzing the effect of the smoothness regularization loss $\mathcal{L}_c$ on color consistency and saturation in 3D reconstructions. The top row shows results without $\mathcal{L}_c$, while the bottom row displays results with $\mathcal{L}_c$ applied. Without the smoothness constraint, the models exhibit noticeable color artifacts, including oversaturated and unnatural textures, as seen in the ***Elephant skull***, where the surface appears overly blue and unevenly colored. Similarly, in the ***Astronaut riding a horse***, the textures are overly contrasted, leading to unrealistic color transitions. In contrast, incorporating $\mathcal{L}_c$ effectively smooths color distributions, producing more natural and visually coherent results. For example, the snail on the leaf demonstrates improved color consistency, reducing abrupt shifts in hue while maintaining fine details. These results indicate that the smoothness loss $\mathcal{L}_c$ is essential for mitigating excessive color saturation and ensuring high-quality, realistic 3D reconstructions.

Further, in Figure 12, we illustrate a comparison with the incorporation of 2DGS (Huang et al., 2024) and demonstrate that incorporation of our proposed regularization terms leads to sharper renderings and improved geometric structures, exhibiting both smoother and more detailed features.

**Pruning ablation.** To evaluate the effectiveness of our densification and pruning strategy, we compare it against two baselines: (1) no densification, and (2) the standard densification approach from the original 3D Gaussian Splatting (3DGS) paper by Kerbl et al. (2023). Table 3 summarizes the results in terms of number of Gaussians and training time. Without any densification and pruning, the method cannot reach satisfactory quality, regardless of the initial number of Gaussians. The standard 3DGS densification produces a much larger number of Gaussians (up to 13–20 million), resulting in significantly longer training times (∼1–1.5

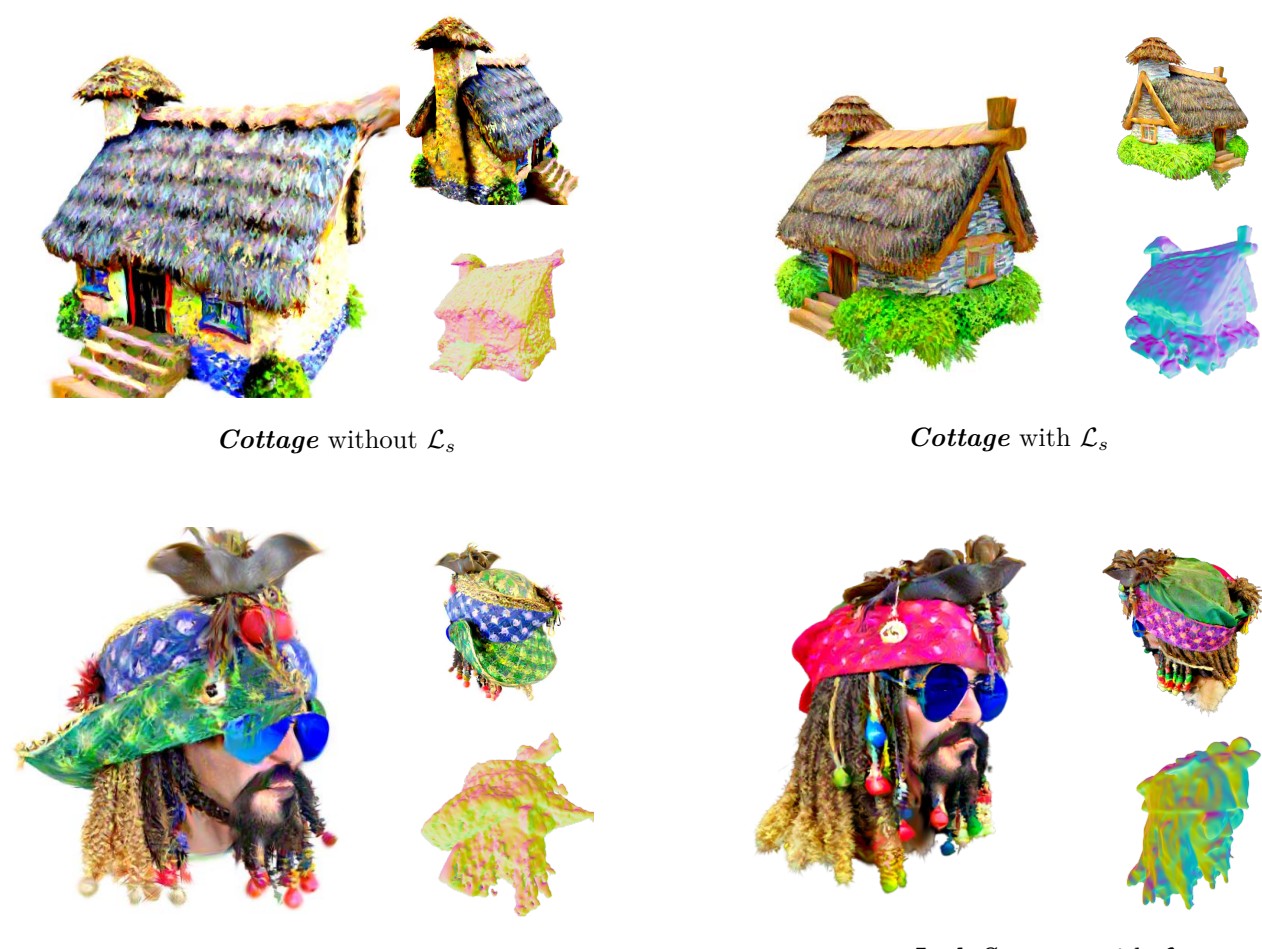

$Cottage$ without $\mathcal{L}_s$                        $Cottage$ with $\mathcal{L}_s$

$Jack\ Sparrow$ without $\mathcal{L}_s$               $Jack\ Sparrow$ with $\mathcal{L}_s$

Figure 9: Qualitative comparison of surface reconstruction on two examples with and without the regularization loss $\mathcal{L}_s$.

hours). In contrast, our method maintains a compact set of Gaussians ($\sim$1M) and converges much faster ($\sim$25 minutes).

Figure 13 shows qualitative results on the prompt *A woolly mammoth*. Without densification, the generated shape remains blurry and lacks detail even with more initial Gaussians. The standard 3DGS densification improves the quality but at the cost of much higher number of Gaussians and slower training. Our method achieves comparable or better quality with fewer Gaussians and faster convergence.

| Method | No. gaussians init | Final no. gaussians | Time to converge |
|---|---|---|---|
| No densification | 50k | 50k | Failed |
| | 100k | 100k | Failed |
| | 1M | 1M | Failed |
| 3DGS densification | 5k | $\sim$13–20M | $\sim$1–1.5 hrs |
| Ours | 5k | $\sim$1M | $\sim$25 mins |

Table 3: Comparison of densification methods.

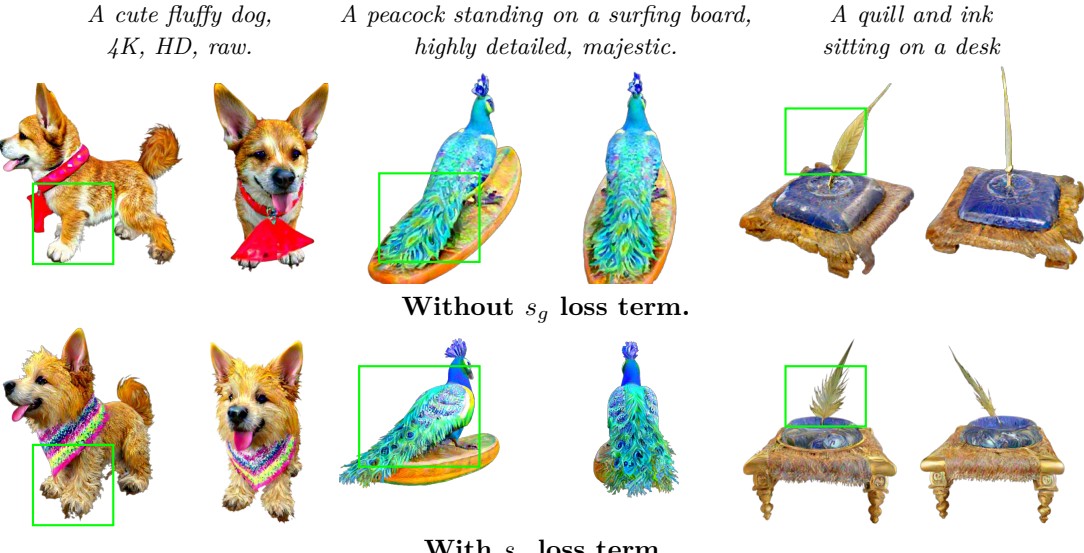

Figure 10: Ablation study on the $s_g$ loss term.

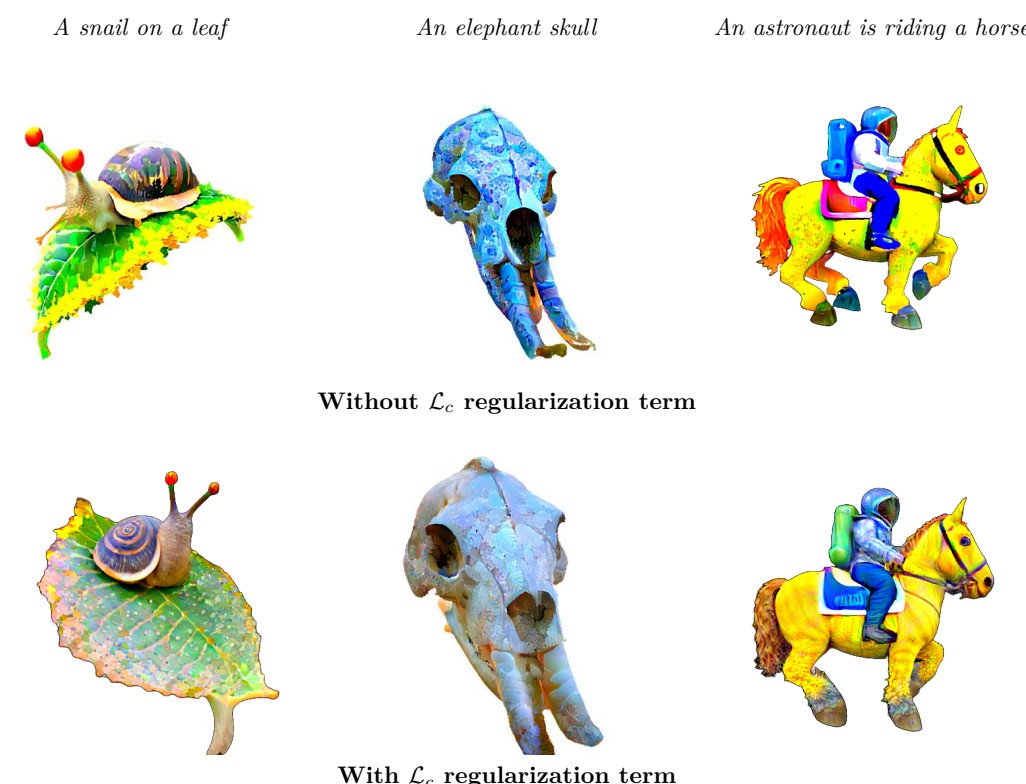

Figure 11: Ablation study on the color and depth smoothness loss.

3DGS                          2DGS

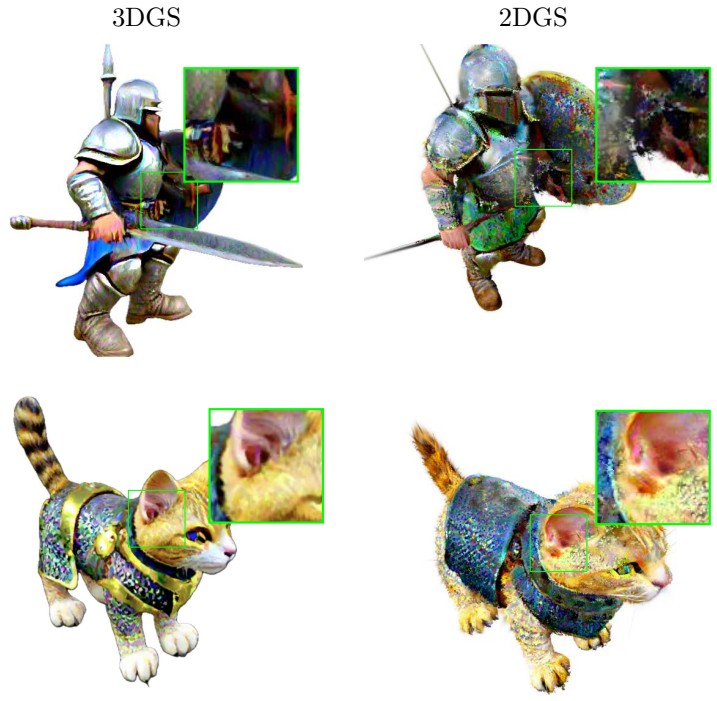

Figure 12: Artifacts observed when densification is based on 2D Gaussian splatting (2DGS) compared to 3D Gaussian splatting (3DGS). The 2DGS method results in noticeable artifacts, particularly around regions like the hilt of the weapon and the tail of the cat, leading to a loss of details.

| | No densification | | | 3DGS | **Ours** |
|---|---|---|---|---|---|
| **Init** | 50k | 100k | 1M | 5k | 5k |
| **Final** | 50k | 100k | 1M | ∼13.4M | ∼0.9M |

Figure 13: Qualitative comparison of densification strategies on the prompt ***A woolly mammoth***. Our method achieves higher visual quality with fewer Gaussians and faster training time, compared to no densification or standard 3DGS densification.

### A.3    More Qualitative Comparisons

Figure 14 presents additional qualitative comparisons of our method, MVGaussian, with GaussianDreamer, LucidDreamer, and GSGEN across various prompts.

For the ***Michelangelo dog statue***, our model accurately captures both the style and the cellphone, while GaussianDreamer and GSGEN miss the cellphone, and LucidDreamer suffers from a multi-face Janus issue. In the ***Steampunk airplane*** prompt, our method integrates the steampunk aesthetic effectively, unlike other methods that produce fighter jets without the steampunk elements. For the ***Opulent couch*** prompt, GaussianDreamer, LucidDreamer, and MVGaussian produce detailed, prompt-aligned models, whereas GS-GEN's output is of lower quality. In the ***Hatsune Miku robot*** prompt, our method captures the anime aesthetics, avoiding the distortions seen in other methods. Finally, in the ***Flamethrower*** prompt, MV-Gaussian, along with GaussianDreamer and LucidDreamer, produces detailed and cohesive models, while GSGEN's result lacks artistic detail and quality.

Overall, MVGaussian outperforms other methods in producing high-quality, detailed, and prompt-aligned 3D models.

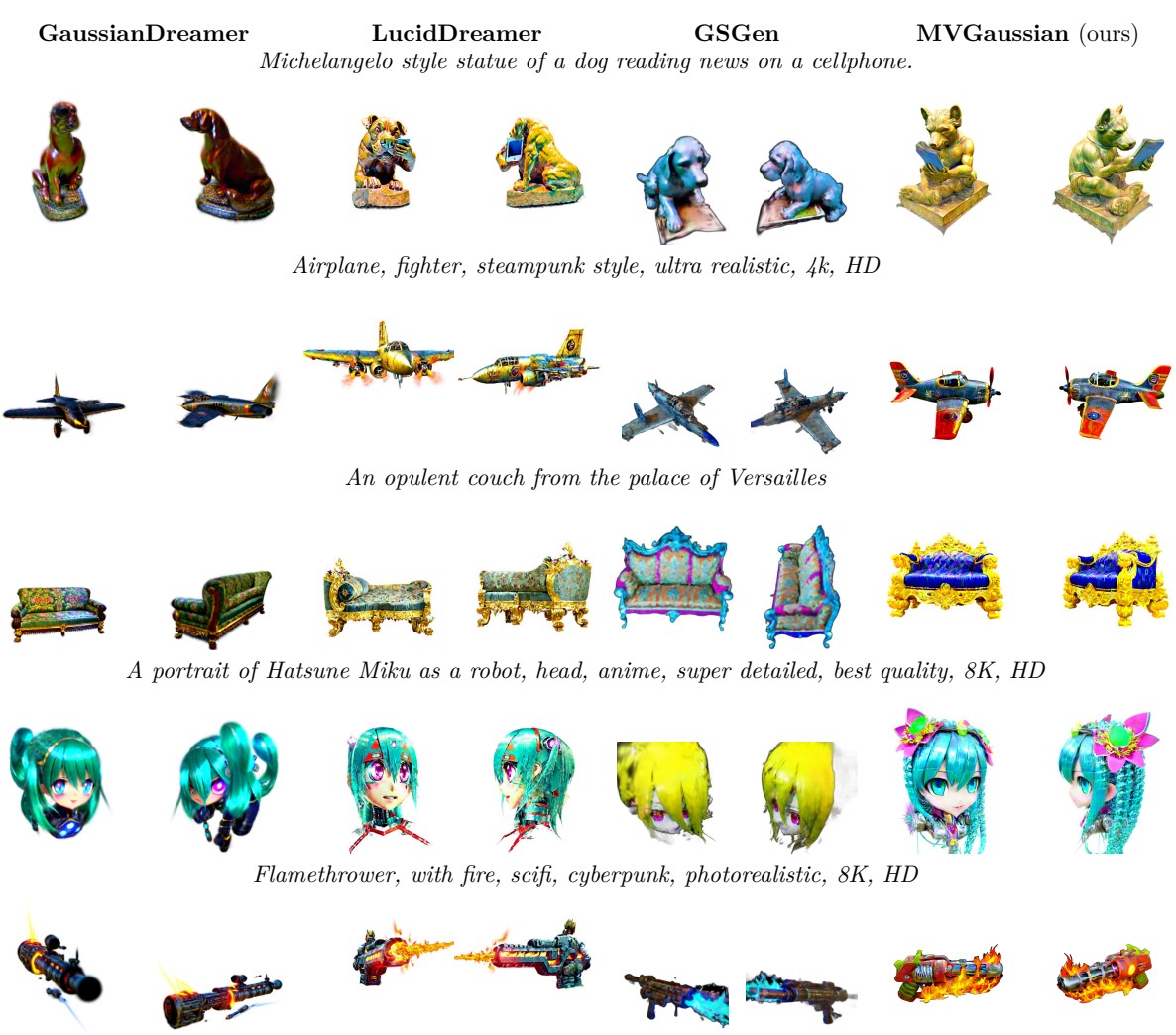

Figure 14: Additional comparison results with existing methods.

**Comparison with MVDream:** Since our method leverages the same multi-view diffusion guidance as MVDream (Shi et al., 2024), we include a direct comparison to better illustrate the effects of our optimization framework. Figure 15 presents qualitative results on several prompts. MVDream requires approximately one hour of training per prompt, yet we observe several limitations in the generated results. For the prompt ***Furry cat***, MVDream produces an irregular, overly long and distorted tail that disrupts the structure of the cat. In the ***Hatsune Miku robot*** prompt, the output lacks any robotic elements and resembles a standard character portrait. In the ***Blue jay*** example, MVDream omits the basket entirely, placing the bird on a set of macarons instead. For the ***Michaelangelo dog statue*** prompt, MVDream generates a statue of a human figure rather than a dog. Finally, for the ***Castle*** prompt, MVDream fails to produce a plausible 3D structure, generating instead a flat cliffside scene. In contrast, our method produces more complete and geometrically consistent 3D results that more closely align with the given prompts.

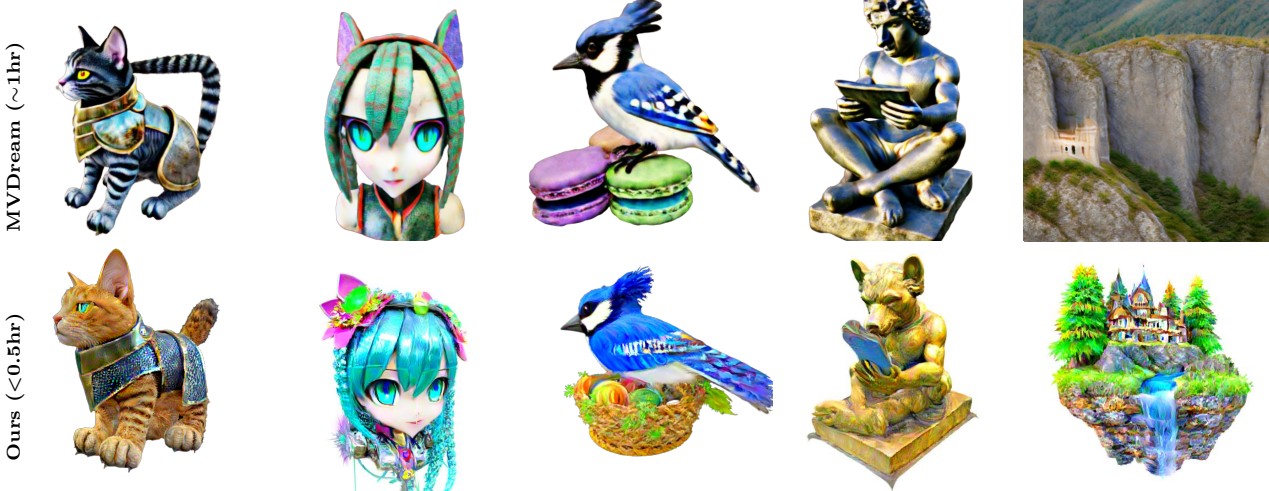

Figure 15: Qualitative comparison with MVDream (Shi et al., 2024) on five prompts: ***Furry cat***, ***Hatsune Miku robot***, ***Blue jay***, ***Michaelangelo dog statue***, and ***Castle***.

### A.4 Implementation Details

**3D Gaussian Splatting (3DGS):** Similar to the original 3DGS implementation (Kerbl et al., 2023), we primarily retain the initial learning rates for positions, color, opacity, scaling and rotation, with adjustments to clip these values to prevent excessively small rates that could hinder convergence. The learning rate ranges are $[1.6 \times 10^{-4}, 1.6 \times 10^{-6}]$ for position, $[3 \times 10^{-3}, 2.5 \times 10^{-3}]$ for color, $[0.1, 0.05]$ for opacity, $[5 \times 10^{-3}, 1 \times 10^{-3}]$ for scaling, and $[1 \times 10^{-3}, 2 \times 10^{-4}]$ for rotation. We initialize 5000 Gaussians and train for 10000 iterations, initiating the densification and pruning process after 1000 iterations and repeating it every 200 iterations. We avoid starting densification too early or too frequently, as this can lead to the creation of redundant Gaussians that complicate optimization. Densification and pruning are halted after 8000 iterations, allowing the final 2000 iterations to focus on optimizing the existing Gaussians.

Densification is performed by cloning or splitting Gaussians with accumulated gradients greater than 0.05, while pruning removes Gaussians with opacity below 0.05. Additionally, we apply a surface pruning technique to eliminate redundant Gaussians, retaining only those near the surface. Algorithm 1 provides pseudo code for our pruning algorithm. For each surface point, we identify the 5 nearest neighbors from the set of Gaussian centers, preserving these close-to-surface Gaussians while pruning those farther away. We also offer an option to retain a percentage $p$ of the remaining Gaussians by calculating the distances from Gaussian centers to the surface and pruning those with distances exceeding the threshold determined by the $p$ percentile.

---

**Algorithm 1: Surface Point Extraction and Pruning**

---

1: **function** GET_SURFACE_POINTS($\mathcal{G}$)
2:     $\mathcal{P} \leftarrow \emptyset$                                                 ▷ Initialize list of surface points
3:     $(\theta, \phi) \leftarrow$ sample_camera_positions()                     ▷ Sample azimuth and elevation angles
4:     **for** each $(\theta_i, \phi_i)$ in $(\theta, \phi)$ **do**
5:         $\mathbf{C} \leftarrow$ get_camera_pose($\theta_i, \phi_i$)                         ▷ Compute camera pose
6:         $\mathcal{I}, \mathcal{D} \leftarrow$ render_views_from_3dgs($\mathbf{C}, \mathcal{G}$)               ▷ Render RGB and depth images
7:         $\mathbf{R}, \mathbf{T} \leftarrow$ get_cam_parameters()                   ▷ Retrieve camera intrinsics/extrinsics
8:         $\mathcal{P}_i \leftarrow$ project_image2world($\mathcal{I}, \mathcal{D}, \mathbf{R}, \mathbf{T}$)         ▷ Back-project image points to world space
9:         $\mathcal{P}_i \leftarrow$ remove_low_density_points($\mathcal{P}_i$)               ▷ Remove unreliable points
10:     **end for**
11:     $\mathcal{P} \leftarrow$ add_points($\mathcal{P}_i$)                         ▷ Aggregate extracted surface points
12:     **return** $\mathcal{P}$
13: **end function**
14: **function** SURFACE_PRUNING($\mathcal{G}, p = 0$)
15:     $\mathcal{P} \leftarrow$ get_surface_points($\mathcal{G}$)                         ▷ Extract surface points
16:     $\mathcal{C} \leftarrow$ get_gaussian_centers($\mathcal{G}$)                     ▷ Retrieve Gaussian centers
17:     **if** $p > 0$ **then**                                       ▷ Use percentile-based pruning if $p > 0$
18:         $\mathcal{D} \leftarrow$ compute_knn_distances($\mathcal{C}, \mathcal{P}, k = 1$)             ▷ Compute nearest surface distances
19:         $\tau \leftarrow$ compute_quantile($\mathcal{D}, p$)                     ▷ Set distance threshold at $p$ percentile
20:         $\mathcal{M} \leftarrow \mathcal{D} > \tau$                         ▷ Mark Gaussians exceeding threshold for pruning
21:     **else**
22:         $\mathcal{I} \leftarrow$ compute_knn_indices($\mathcal{P}, \mathcal{C}, k = 5$)                   ▷ Find nearest Gaussian indices
23:         $\mathcal{M} \leftarrow$ compute_pruning_mask($\mathcal{I}$)                     ▷ Determine pruning mask
24:     **end if**
25:     $\mathcal{G}' \leftarrow$ prune_gaussians($\mathcal{C}, \mathcal{M}$)                       ▷ Remove pruned Gaussians
26:     **return** $\mathcal{G}'$                                     ▷ Return remaining Gaussians
27: **end function**

---

**Multi-view Guidance:** We utilize the pretrained model (*sd-v2.1-base-4view*) provided by MVDream ([Shi et al., 2024](#)), which is based on the diffusion checkpoint at *stabilityai/stable-diffusion-2-1-base*. To adapt MVDream's guidance, we generate four views that are linearly distributed around the object at a random elevation. These four views are then used to compute the SDS loss as introduced in DreamFusion ([Poole et al., 2022](#)). We also use the following negative prompt to guide the generation model: *"shadow, oversaturated, low quality, unrealistic, ugly, bad anatomy, blurry, pixelated, obscure, unnatural colors, poor lighting, dull, unclear, cropped, lowres, low quality, artifacts, duplicate, morbid, mutilated, poorly drawn face, deformed, dehydrated, bad proportions."*

The overall algorithm of our approach is outlined in Algorithm [2](#). It involves iteratively optimizing Gaussian parameters with integrated densification and pruning. The process includes rendering multiple views, computing losses, and refining the model by selectively densify and prune existing Gaussians.

**Hardware Setup:** Our experiments were run on a system equipped with an NVIDIA A100 Tensor Core GPU with 80 GB of VRAM, supported by two AMD EPYC 7543 32-Core Processors. The system supports a total of 128 CPUs.

---

**Algorithm 2: Overall Optimization Process**

---

 1: **function** OPTIMIZE_GAUSSIANS($\mathcal{G}, N$)
 2:     Initialize optimizers for Gaussian parameters and loss weights
 3:     **for** each $i \in \{1, \dots, N\}$ **do**
 4:         $\Theta \leftarrow$ sample_camera_positions(4)                    ▷ Randomize four camera positions
 5:         $\mathcal{B} \leftarrow$ randomize_background_color()
 6:         $\mathcal{I}, \mathcal{D} \leftarrow$ render_views_from_3dgs($\Theta, \mathcal{G}, \mathcal{B}$)
 7:         $\mathcal{L}_{\mathrm{SDS}} \leftarrow$ compute_sds_loss($\mathcal{I}, \mathcal{D}$)
 8:         $\mathcal{L}_{\mathrm{reg}} \leftarrow$ compute_regularization_losses($\mathcal{G}$)
 9:         $\mathcal{L} \leftarrow$ compute_final_weighted_loss($\mathcal{L}_{\mathrm{SDS}}, \mathcal{L}_{\mathrm{reg}}$)
10:         perform_backpropagation($\mathcal{L}$)
11:         **if** check_densification_requirement() **then**
12:             $\mathcal{P} \leftarrow$ get_surface_points($\mathcal{G}$)
13:             $\mathcal{G} \leftarrow$ surface_pruning($\mathcal{G}$)
14:             **if** compute_gaussian_distance($\mathcal{G}, \mathcal{P}$) $> \tau$ **then**
15:                 remove_gaussians($\mathcal{G}$)
16:             **end if**
17:             **if** check_opacity_reset_requirement() **then**
18:                 reset_opacity($\mathcal{G}$)
19:             **end if**
20:         **end if**
21:         perform_optimization_step($\mathcal{G}$)
22:     **end for**
23:     **return** $\mathcal{G}$
24: **end function**

---

### A.5  Parameter analysis

We evaluate the sensitivity of $w_s$ and $w_c$ by running each configuration on 8 prompts listed in Table 1. The loss weights are relative, so we fix $w_{\mathrm{sds}} = 1$ and vary only $w_s$ and $w_c$ to understand their effects. We report four metrics: **CLIP score** (range 0-1), which measures alignment between the generated image and the text prompt; **ImageReward** (Xu et al., 2023), a continuous score estimating image quality and relevance; **ChatGPT-4o** score, where we prompt ChatGPT-4o to rate the image from 1 to 10 based on how well the generated object matches the input prompt in terms of object shape, geometry, texture, and overall quality; and **Human preference**, based on pairwise comparisons. For all metrics, higher scores indicate better quality. As shown in Tables 4 and 5, extreme values of $w_s$ or $w_c$ lead to lower quality and alignment, while moderate values ( $w_s = 200$, $w_c = 200$) often yield the better results. We also observe that while automatic metrics provide useful guidance, they do not always align with human judgment, underscoring the importance of incorporating human evaluation in tuning these parameters.

| $w_s$ | CLIP | ImageReward | ChatGPT 4o | Human preference |
|-------|------|-------------|------------|------------------|
| 1     | 0.21 | -0.15       | 5.86       | ✗                |
| 50    | 0.18 | 0.16        | 5.92       | ✗                |
| 100   | **0.26** | 0.32    | 6.54       | ✗                |
| 200   | 0.25 | **0.33**    | **7.83**   | ✓                |
| 400   | 0.22 | 0.11        | 6.44       | ✗                |

Table 4: Sensitivity analysis of surface regularization weight $w_s$.

Figure 16 illustrates the effect of the pruning threshold $\tau$, which controls the maximum distance allowed between a Gaussian and the estimated pseudo-surface. As shown with the ***Bichon frise*** prompt, a smaller $\tau$ leads to more aggressive pruning. For example, at $\tau = 0.001$, only ∼0.7M Gaussians remain, but the model loses many fine details and the overall surface quality degrades. In contrast, a large $\tau$ such as 0.05 retains more Gaussians (∼2M), but results in slower optimization and diminishing returns in visual quality.

| $w_c$ | CLIP | ImageReward | ChatGPT 4o | Human preference |
|---|---|---|---|---|
| 1 | 0.20 | 0.10 | 6.21 | ✗ |
| 50 | 0.27 | 0.15 | 7.34 | ✗ |
| 100 | 0.28 | **0.35** | **7.50** | ✗ |
| 200 | **0.31** | **0.35** | 7.45 | ✓ |
| 400 | 0.21 | -0.05 | 5.33 | ✗ |

Table 5: Sensitivity analysis of the smoothness regularization weight $w_c$.

$\tau = 0.001$ $\qquad\qquad\qquad$ $\tau = 0.02$ $\qquad\qquad\qquad$ $\tau = 0.05$

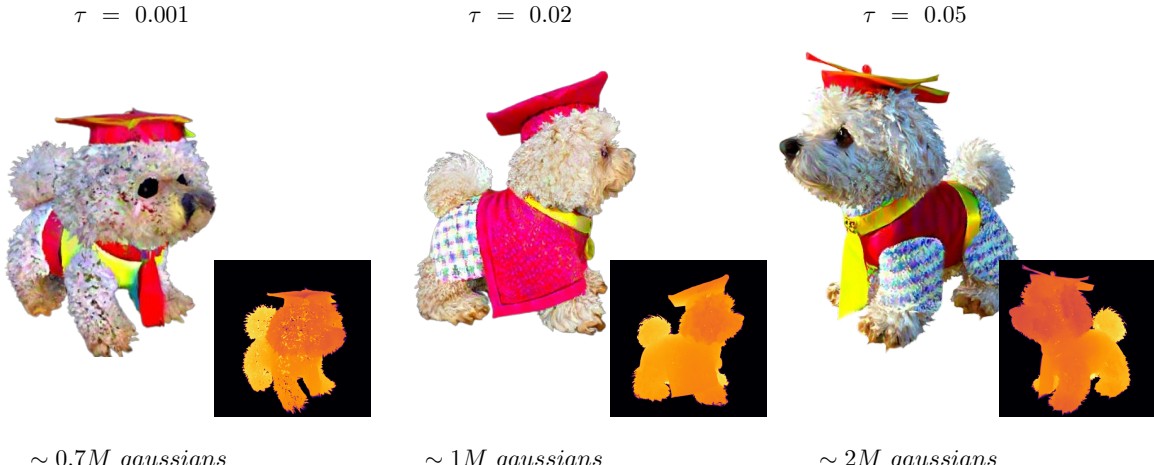

$\sim 0.7M\ gaussians$ $\qquad\qquad$ $\sim 1M\ gaussians$ $\qquad\qquad$ $\sim 2M\ gaussians$

Figure 16: Effect of pruning threshold $\tau$ on the prompt ***"a bichon frise wearing academic regalia, 8K, HD, raw."***.

We find that $\tau = 0.02$ provides a good balance between quality and efficiency, yielding compact Gaussian sets (∼1M) while maintaining good surface fidelity and fast training.

## A.6   Additional Results

Figure 17 showcases additional results generated by our model using a generic prompt template such as ***"a DSLR photo of a …"***. The diverse range of objects includes a castle, a cyborg, a marble bust of Captain America, a dragon, Gandalf, and so on. These examples demonstrate the model's capability to create highly detailed and visually coherent 3D representations across various subjects, from complex fantasy characters to everyday objects, illustrating the robustness and versatility of our approach.

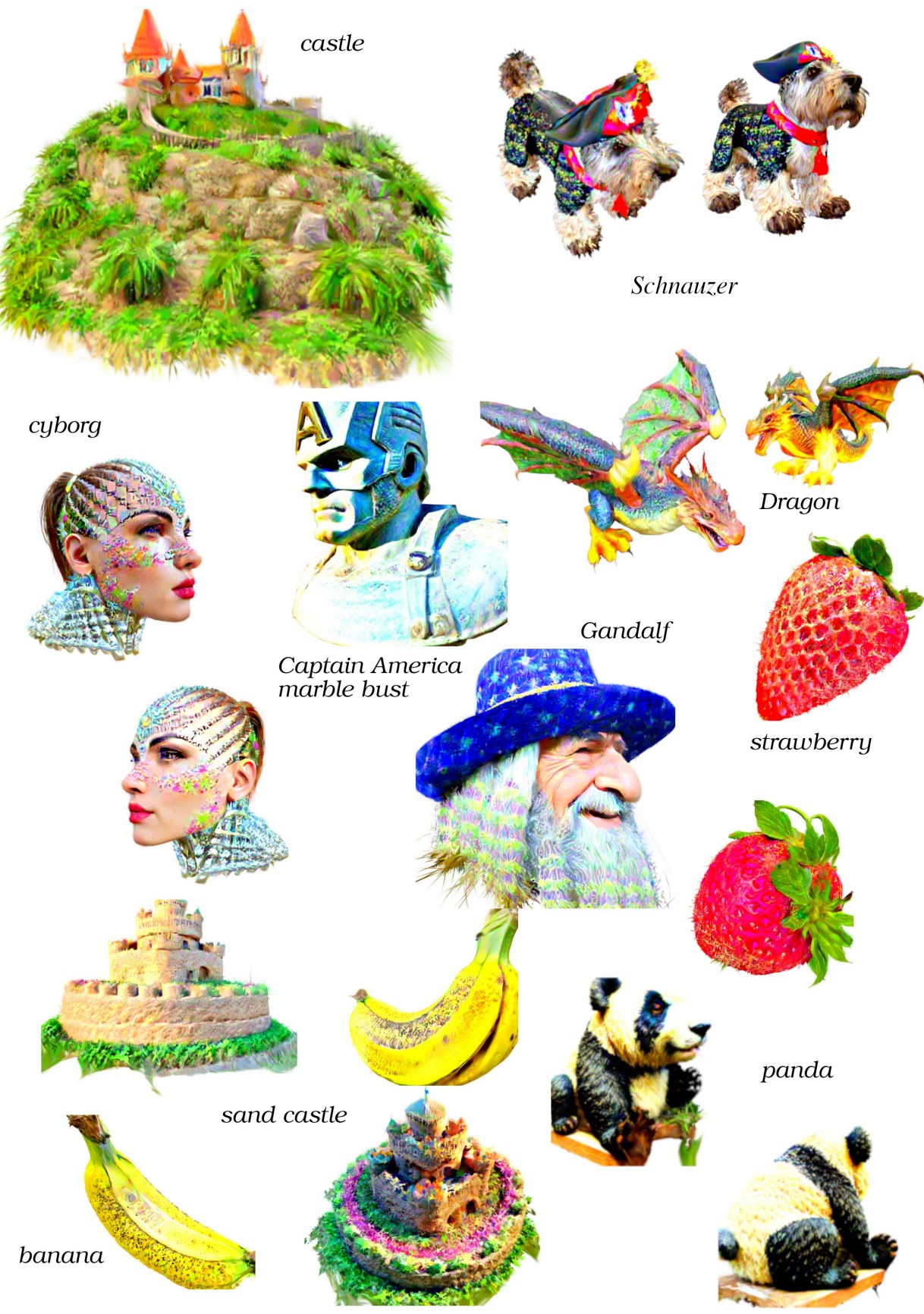

Figure 17: Additional results generated using generic prompt template of "a DSLR photo of a ...".

