# OpenReview forum: "MVGaussian: High-Fidelity text-to-3D Content Generation with Multi-View Guidance and Surface Densification"
_TMLR — Rejected by TMLR_

### Review · Reviewer_5zJK · 2025-06-09

**Summary Of Contributions:**

This paper proposes a method to generate 3D Gaussian Splatting (3DGS) assets from text input using the Score Distillation Sampling (SDS) paradigm. The method consists of three major components: 1. the use of a diffusion prior from MVDream that incorporates multi-view information; 2. a 3DGS regularizer that encourages the Gaussians to be close to the surface and be flat; 3. a method to densify and prune Gaussians based on their proximity to the depth surface.

**Audience:**

Yes

**Broader Impact Concerns:**

Not that I am aware of.

**Claims And Evidence:**

No

**Requested Changes:**

The requested changes are implied in the weaknesses section. Among them, "Clarity of writing" is a minor factor. All other points are critial to the paper.

**Strengths And Weaknesses:**

Strength: the densification and pruning methods for Gaussians seem to be a novel approach.

Weaknesses
* In the discussion on recent feedforward methods for 3D generation (e.g. LRM, TRELLIS), the last paragraph of Sec 2.3 writes: "However, despite these improvements, existing 3D generation methods still struggle with consistency, speed, and quality. To address these challenges, we propose our method, MVGaussian, which mitigates the aforementioned limitations while improving efficiency, quality, and robustness in 3D generation." It sounds like the proposed method achieves better 3D consistency and speed compared with these methods. However, this paper does not provide a formal comparison with these feedforward methods. In fact, recent feedforward methods, especially TRELLIS, have achieved significant progress in 3D consistency, and they have always been known for their speed compared with SDS-based methods. I am NOT implying that this method must be compared with TRELLIS or LRM to be accepted, because they belong to different categories of methods; however, if there is such a claim in the paper, the comparison must be provided to justify the claim.
* At the end of Sec. 1, the paper writes, "We introduce a unified framework for text-to-3D content generation that integrates SDS loss with 3D Gaussian splatting with a novel backbone reducing the Janus problem." If the "backbone" refers to the network architecture of the multi-view diffusion prior according to its usual meaning, this claim is also untrue, because, as explained in Sec. 4.1, the method uses the diffusion model from MVDream, an existing network. Even if we take into account the possibility that the method may be the first to use a multi-view aware diffusion prior with 3DGS generation, this contribution is also weak because it is the direct adoption of an existing model.
* Wrong positioning of Gaussian-surface alignments: in the beginning of Sec. 4, the paper writes: "However, unlike SuGaR, instead of a post-processing term, we introduce a novel regularization term that allows for flattening the Gaussians during the learning process itself" and "to the best of our knowledge, this is the first approach that aims to optimize the Gaussians using the estimated depth." Both claims are untrue. The first sentence implies that the regularization in SuGaR is a post-processing method after 3DGS training is done. However, this is not true. Sec 5.1 of SuGaR writes very clearly that "we perform 6,000 iterations with the regularization term introduced in Subsection 4.1". The second sentence is also untrue because SuGaR exactly uses estimated depth for optimization. For example, the caption of Figure 5 in SuGaR writes, "we render depth maps of the Gaussians, sample points p in the viewpoint according to the distribution of the Gaussians. Value f(p) is taken as the 3D distance between p and the intersection between the line of sight for p and the depth map."
* Even if we put aside these factual errors, the contribution ofthe  3DGS regularizer is also not thoroughly verified and ablated. Eq. (11) consists of two terms. The first term is actually a special case of Eq. (8) of SuGaR, when the points are only sampled from the surface, and thus \hat f(p) = 0. Only the second term |s_p| seems to be novel. However, the contribution of this term is not ablated. The ablation study in Sec A.2 only studies the effect of L_s. It is hard to tell the exact contribution of this novel 2nd term. In fact, even the difference of results with and without L_s in Figure 7 is not that obvious. In addition, the paper does not show the extracted surface from the generated 3DGS, so it does not demonstrate the effects of this term on surface quality.
* Clarity of writing: in Eq. (11) it is not clear what the range of points x are sampled from. From the context, we can guess x iterates over the rendered depth maps from the 3DGS, but this information is missing from the math notation. Also, Eq. (12) overloads the use of x for pixels.
* The evaluation lacks the following items:
    * Comparison with MVDream: MVDream uses the exact same multi-view diffusion prior. This paper claims to reduce the Janus effects. Is it really better than MVDream?
    * Recent text-to-3DGS generation methods: e.g., "Connecting Consistency Distillation to Score Distillation for Text-to-3D Generation". Li et al. ECCV 2024. Is the proposed method better than these existing approaches?
    * Incomplete ablation studies: the paper claims "a novel densification method by optimizing the placement and density of Gaussian elements that accelerate the generation process reducing the overall training time to ∼ 25 minutes." However, no ablation studies are provided to demonstrate its effect. It should at least show that it leads to faster training compared with 1. no densification and pruning at all; 2. standard densification and pruning from the original 3DGS paper.

---

> ### Author Response · Authors · 2025-06-25
> **Response to reviewer 5zJK's concerns about the writing, the contribution and ablation studies**
>
> We thank reviewer **5zJK** for the detailed review and thoughtful feedback. We address the main points raised below, following the order in which they appeared.:
> 1) We agree that LRM and TRELLIS are fundamentally different from SDS-based methods and have made impressive progress in speed and consistency. Our statement in Sec. 2.3 was intended to reference prior SDS-based pipelines, not to claim superiority over feedforward approaches like TRELLIS. We apologize for the ambiguous phrasing and have revised the paragraph to clarify this scope and avoid overgeneralization. To maintain focus and fair comparisons within the same class of methods, we chose to benchmark against SDS-based approaches using 3DGS.
> 2. The term "novel backbone" was not intended to suggest that we modified the architecture of MVDream. Rather, we meant to emphasize the novel integration of multi-view priors with a customized 3DGS optimization framework. We revised the text to:
> "We introduce a text-to-3D framework that combines 3D Gaussian Splatting with multi-view diffusion guidance and novel regularization terms, enabling more efficient optimization with reduced Janus artifacts."
> 3. Upon revisiting the SuGaR paper, we acknowledge that regularization is applied after 9k iterations of initial training (7k for positions and 2k for opacity). We mistakenly mentioned the later stage as a post-processing stage. We also confirm that it does leverage estimated depth for the later stage of the optimization.
> To clarify, our novelty lies in (a) using backprojected depth directly within the densification and pruning loop (Algorithm 1), and (b) formulating a simplified and lightweight regularization loss that avoids precomputed SDFs and integrates seamlessly with SDS optimization. We acknowledge the points put forth by the reviewers and have make the necessary corrections to prevent the mischaracterization and revised the phrasing to emphasize the technical differences.
> 4. We agree that our regularizer draws inspiration from SuGaR's surface alignment term. However, our formulation is intentionally designed to be simpler and more efficient for SDS-based text-to-3D generation. In particular, our implementation does not require computing or regressing SDF values, but instead operates directly on surface samples derived from rendered depth maps. This makes it more intuitive and cheaper to integrate into the optimization loop, which supports our goal to maintain fast convergence within an SDS framework.
> To address the reviewer's suggestion, we have now added an explicit ablation isolating the effect of the flattening term $|s_g|$ in Appendix Section A.2 (Figure 10), and we provide surface reconstruction visualizations (Figure 9) to better show the contribution of $L_s$ to geometric quality. We hope this clarifies the contribution of the proposed regularization terms.
> 5. We fixed the overloaded notation in Eqs. 12, 13, and 15. We explicitly stated that the points x in Eq. (11) are sampled from the pseudo surface constructed from the rendered depth maps.
> 6. 1) **Comparison with MVDream**: We have added a direct qualitative comparison with MVDream in Appendix Section A.3 (Figure 15). Using the same multi-view diffusion prior, we observe that our method produces more geometrically consistent and complete 3D results, particularly on prompts where MVDream suffers structural distortions, or missing objects, or even complete failure. Additionally, MVDream requires $\sim$ 1hr of training per prompt, whereas our method achieves comparable or better quality in $\sim$25 minutes. We hope this comparison clarifies the strengths of our approach.
>     2) **Comparison with GCS**: The referenced method (GCS) shares similarities with recent 3DGS-based pipelines, such as LucidDreamer (Liang et al., 2023), and also relies on Shap-E/Point-E initialization. We have included this method in our comparison with the main baselines (Figures 3 and 4). The rendered videos provide the best illustration of the typical weaknesses of these models, such as inconsistencies, geometric artifacts, and Janus effects. We encourage the reviewer to view the videos provided on our project page and in the supplementary materials for a more comprehensive comparison.
>    3) **Ablation on densification strategy**: We have added an explicit ablation study of the densification and pruning strategy, comparing our method against: (1) no densification, and (2) the standard densification approach from the original 3D Gaussian Splatting (3DGS) paper. The results are presented in Appendix Section A.2: Pruning ablation (Table 3 and Figure 13). As shown, the model failed to converge without densification, regardless of the number of initial Gaussians. Our method achieves faster convergence, reducing training time by 2–3$\times$ compared to standard 3DGS densification, while also producing fewer redundant Gaussians and maintaining better surface quality.

---

> > ### Comment · Reviewer_5zJK · 2025-06-29
> >
> > First, I appreciate the efforts by the authors to revise the paper and provide more experiments in the rebuttal.
> >
> > Second, I would like to add a comment to the authors' response that "our implementation does not require computing or regressing SDF values, but instead operates directly on surface samples derived from rendered depth maps". However, the SDF values in SuGAR are not real ground truth SDF either. They are similar to the approach in this paper, computed as offsets from the rendered depth map (as shown in Figure 5 in SuGAR). Thus, the complexity in computing the surface regularization sounds similar between these two works.
> >
> > Third, upon looking at Figure 10, I cannot observe noticeable differences between results with and without $|s_g|$ loss term, which is the real contribution by this paper in Eq. (11). The other loss term in Eq. (11), as previously pointed out, is a special case of Eq. (8) in SuGAR.
> >
> > I would like raise the attention of the action editors to these points when making the final decision.

---

> > > ### Author Response · Authors · 2025-06-30
> > >
> > > We thank reviewer **5zJK** for the follow-up.
> > >
> > > Regarding the $\mathcal{L}_s$ loss term, our formulation avoids the explicit local surface distance matching and additional **normal alignment** term used in SuGaR (their Equations 8 and 10). Instead, we directly penalize distances along the smallest Gaussian axis. Beyond the $\mathcal{L}_s$ term, our contributions include the $\mathcal{L}_c$ color and depth regularization, as well as a surface-guided pruning scheme. The $\mathcal{L}_s$ loss helps push Gaussians toward the surface, which in turn enables more effective pruning and reduces redundant computation. Together, these components form an integrated pipeline that achieves higher quality and faster training compared to existing 3DGS baselines.
> > >
> > > While the differences from $|s_g|$ alone may appear subtle in static images (Figure 10), the benefits are clearer in multi-view rendered videos (included in the **Supplementary Material**), which show reduced thickening and more consistent delicate and thin structures such as fur and feathers, resulting in better prompt alignment. If the reviewer has not yet viewed the videos, we encourage them to do so.

---

### Review · Reviewer_XAeV · 2025-06-11

**Summary Of Contributions:**

This paper introduces MVGaussian, a unified framework for high-fidelity text-to-3D content generation. The authors aim to address two critical gaps in existing methodologies: the "Janus problem" (multi-face ambiguities) caused by imprecise Score Distillation Sampling (SDS) guidance, and the largely unexplored optimization of 3D Gaussian Splatting for representing 3D volumes. Extensive experiments are claimed to validate the approach, demonstrating that it produces high-quality visual outputs with minimal time cost.

**Audience:**

Yes

**Broader Impact Concerns:**

No further comments

**Claims And Evidence:**

Yes

**Requested Changes:**

See weaknesses.

**Strengths And Weaknesses:**

### Strengths

- The paper directly confronts the prevalent "Janus problem" (multi-face ambiguities) in text-to-3D generation by proposing multi-view guidance, which is a crucial step towards more consistent and accurate 3D model generation.
- The claim of achieving high-quality results within "half an hour of training," offering a "substantial efficiency gain over most existing methods," is a major strength. Faster training times are critical for practical applications and iterative design.

### Weaknesses

- The structure of the proposed pipeline is straightforward on the 3D-GS representation, which is not totally new for me, mainly similar to the Magic3D. The new points are summarized in some loss terms and depth maps. From this side, I think the novelty is not significant for the TMLR. Other than that, there are no extra ablations on the depth maps to demonstrate the effectiveness. For the ablations in Figure 7, the improvements of L_s are not clear for these presented examples. The generated objects by w. L_s or w.o. L_s also seems good for me.
- Compared to the previous nerf-based representation, the 3D-GS is very efficient. The minimal time cost on generated objects does not come from the newly proposed points.
- I am very curious about the normal map for the generated objects since the proposed method introduces the surface densification and pruning via depth maps.

- While the densification algorithm aligns Gaussians "close to the surface," the extent to which it handles complex geometries, thin structures, or highly textured surfaces is not detailed. How does MVGaussian perform when generating 3D content from more complex or abstract text prompts, or for objects with intricate details, open structures, or transparency?
- The claim of efficiency gain over "most existing methods" is broad. A more specific mention of which methods are outperformed and by what margin would strengthen the argument. Some recent work should be considered and discussed:
- [1] Hallo3d: Multi-modal hallucination detection and mitigation for consistent 3d content generation
- [2] Gaussianobject: High-quality 3d object reconstruction from four views with Gaussian splatting
- [3] ESCT3D: Efficient and Selectively Controllable Text-Driven 3D Content Generation with Gaussian Splatting
- Can MVGaussian be extended to generate entire 3D scenes (not just individual objects) from text prompts? What are the authors' plans for future work, particularly concerning animation, interactivity, or integration into larger 3D environments?

---

> ### Author Response · Authors · 2025-06-25
> **Response to reviewer XAeV's concerns about our contribution, ablation studies and writing**
>
> We thank reviewer XAeV for the thorough review and feedback. Below, we respond to the main points in the order they were raised.
>
> 1. - Our pipeline is completely different from Magic3D in several key aspects: (1) Magic3D is a NeRF-based, two-stage (coarse-to-fine) approach, whereas ours is a 3D Gaussian Splatting-based single-stage pipeline; (2) Magic3D’s performance gains largely come from combining existing techniques such as Instant NGP and DMTet, and its higher mesh resolution depends on fine-tuning a latent diffusion model. In contrast, our performance improvements stem from our novel loss design and an intuitive surface-based pruning strategy, which directly reduces redundant computation; (3) Magic3D has already been extensively compared to 3DGS-based baselines used in our work, and prior results have shown that it performs worse than these baselines. Our method outperforms all 3DGS baselines both in terms of visual quality and speed.
>     - Regarding the ablation studies, we have added additional experiments analyzing the effect of the $s_g$ flattening term and our pruning strategy (see Appendix A.2, Figure 10, Table 3, and Figure 13), as well as more surface reconstruction results (Appendix A.2, Figure 9). For Figure 7 (now becomes Figure 8 in our revised manuscript), we have updated the examples to better highlight areas where our method provides improvements. We also encourage the reviewer to visit our project page or refer to the video results in the supplementary, where the differences are more apparent.
> 2. - We agree that 3D Gaussian Splatting (3DGS) is fundamentally more efficient than NeRF-based representations, and this is a key motivation for adopting 3DGS in our work. However, our contribution goes beyond simply using 3DGS: our loss design and surface-based pruning strategy significantly improve training efficiency and reduce redundant computation within the 3DGS framework. Existing 3DGS-based text-to-3D approaches still require substantial time (Figure 5).
>    - In contrast to other methods, our method reduces training time to $\sim$25 minutes, while achieving higher visual quality. As shown in our densification ablation (Appendix A.2, \textbf{Pruning ablation}), our approach reduces training time by 2–3$\times$ compared to standard 3DGS densification, and produces more compact Gaussian sets ($\sim$1M vs. $\sim$13–20M Gaussians).
> 3. We have added the surface normal for the main prompts in Figure 3 and Figure 4 of the main manuscript. We've also included surface normal in the Figure 9 (Appendix A.2) of our ablation study for the effect of the $L_s$ loss term.
> 4. - We demonstrate the capability of our model to generate intricate details using several open-ended prompts, such as **Castle** and **Orc**. We also showcase free-form object generation (e.g., **Flamethrower**, **Castle**), as entities like flames and waterfalls are inherently difficult to model due to their lack of definitiveness.
>    - Additionally, we have expanded our results to better illustrate the behavior of our method on thin structures (**Quill and ink** prompt in Figure 10) and transparent objects (**Wine glass** prompt in Figure 8). Overall, our surface-based pruning and regularization improve fidelity for thin structures and intricate details. However, modelling fully transparent objects remain challenging, due in part to limitations in the diffusion prior and the inherent difficulty of handling opacity in 3DGS optimization. We will further explore these limitations in future work.
> 5. We have clarified the claim by specifying that our method achieves 2$\times$ faster training compared to 3DGS-based approaches such as GSGen and LucidDreamer, while producing higher visual quality. We avoid broad comparisons to methods outside this scope, and have reworded the abstract to more precisely reflect our contributions.
> 6. We have discussed these methods in our \textbf{Related work}, section 2.3. Unfortunately, a direct comparison is not feasible at this time: [1] Hallo3D has not released code, so we could not perform a direct comparison; [2] GaussianObject is an image-to-3D method, whereas our work focuses on text-to-3D generation, making the comparison not directly applicable; [3] ESCT3D is a recent paper and, to our knowledge, code is not yet available for comparison.

---

> > ### Author Response · Authors · 2025-06-25
> >
> > 7. Scene-level generation remains one of the major challenges in text-to-3D content generation, particularly for SDS-based methods. Existing scene-level pipelines typically require an explicit guided layout or strong spatial priors, as studied in Set-the-scene [1], Gala3D [2], and RealmDreamer [3]. Our current work focuses on object-level generation, which we consider an important first step toward more complex scene synthesis. Extending MVGaussian to full scene generation is an exciting direction for future work, but is out of scope for this paper. That said, we do show some examples where our method can handle partial scene or multi-object prompts, such as **Castle high up in the mountains** (Figure 3) and **A blue jay sitting on a willow basket of macarons** (Figure 4), which combine multiple elements. We also plan to explore future extensions toward mesh extraction and animation for other downstream tasks.
> >
> > [1] Cohen-Bar, D., Richardson, E., Metzer, G., Giryes, R. and Cohen-Or, D., 2023. Set-the-scene: Global-local training for generating controllable nerf scenes. In Proceedings of the IEEE/CVF International Conference on Computer Vision (pp. 2920-2929).
> >
> > [2] Xiaoyu Zhou, Xingjian Ran, Yajiao Xiong, Jinlin He, Zhiwei Lin, Yongtao Wang, Deqing Sun, and Ming-Hsuan Yang. 2024. GALA3D: towards text-to-3D complex scene generation via layout-guided generative gaussian splatting. In Proceedings of the 41st International Conference on Machine Learning (ICML'24), Vol. 235. JMLR.org, Article 2570, 62108–62118.
> >
> > [3] Shriram, J., Trevithick, A., Liu, L. and Ramamoorthi, R., 2024. Realmdreamer: Text-driven 3d scene generation with inpainting and depth diffusion. arXiv preprint arXiv:2404.07199.

---

### Review · Reviewer_7fJp · 2025-06-15

**Summary Of Contributions:**

The authors propose MVGaussian, a text to 3D generation framework that combines multiview diffusion based  guidance with an explicit 3D gaussian splatting representation to tackle the well known Janus problem. The incorporation of an on-the-fly regularization scheme is clever and the qualitative comparisons showcase higher visual quality than some well known approaches (DreamGaussian, LucidDreamer).

**Audience:**

Yes

**Broader Impact Concerns:**

MVGaussian enables efficient and photorealistic generation of 3D assets from text inputs, which is valuable for creative and prototyping tasks. However, there is potential for misuse in the form of unlicensed and nefarious content generation. The current manuscript does not fully articulate these concerns and the authors should consider adding these.

**Claims And Evidence:**

Yes

**Requested Changes:**

1. Add a sensitivity analysis for the regularization weights and surface threshold. This will strengthen claims of robustness - critical for acceptance

**Strengths And Weaknesses:**

Strengths:
1. The contributions are well motivated and are presented in a clear manner
2. The ablations, user study and failure analysis provide convincing support
3. Low training time

Weaknesses:
1.  Evaluation focuses on stylized single-object prompts; no complex indoor scenes or multi-object compositions are showcased
2. There is no sensitivity analysis on the chosen threshold parameter for flattening and densification

---

> ### Author Response · Authors · 2025-06-25
> **Response to reviewer 7fJp's concerns about sensitivity analysis**
>
> We thank reviewer **7fJp** for the constructive reviews and feedback. We address the main points below.
>
> 1. **Scene level or multi-object composition evaluation**: Scene-level generation remains a major challenge in text-to-3D content synthesis, especially for SDS-based pipelines, which typically require explicit spatial layouts or strong priors, as explored in Set-the-Scene [1], Gala3D [2], and RealmDreamer [3]. Our current work focuses on object-level generation, which we view as a foundational step toward more complex scene synthesis. While full-scene generation is beyond the scope of this paper, we have added several multi-object and partial scene examples—such as **Castle high up in the mountains** (scene level) and **A blue jay sitting on a willow basket of macarons**, **An armored green-skin orc warrior riding a vicious hog** (multi-object composition) in Figures 3, 4, and the supplementary materials, to highlight the potential of our method in handling more structured or abstract prompts.
>
> 2. **Sensitivity analysis**: We have now added experiments on the pruning threshold $\tau$ (Appendix A.5, Figure 16) and on the flattening term weights $w_s$ and $w_c$ (Tables 4 and 5), showing how these parameters affect quality, geometry, and efficiency. We select the best hyperparameters based on several automatic metrics and human evaluation. We do not ablate $w_{\text{sds}}$ since it serves as the primary supervision signal for generation. Instead, we fix it and vary the auxiliary weights to better understand their individual contributions.
>
> 3. **Safety concern about potential misuse**: We have added a section on ethical concerns and societal impact (Section 9), which discusses potential risks of misuse. We encourage responsible use of this technology and adherence to ethical standards in its deployment.

---

### Decision · Action_Editor_nm28 · 2025-07-28

**Recommendation:** Reject

**Additional Comments:**

The paper was reviewed by 3 experts and received mixed reviews. Two of the reviews pointed out several key issues. The biggest concern was that many of the claimed contributions of the paper were either subsumed by, or elementary derivations of, techniques proposed in earlier papers, notably SUGAR (Guedon and Lepetit, CVPR 2024). There were also many missing comparisons, leaving the evaluation section incomplete. The authors provided lengthy responses to many points addressed above.

After the responses, the average sentiment among the reviewers continued to be negative. I agree with the reviewers. Many claims in the paper need to be tightened before it can be published, and since this is a fast-moving area of research, up-to-date comparisons with the latest methods would be beneficial. Moreover, it can help if the authors provide convincing evidence that the proposed innovations (above those already implied by SUGAR) are actually useful both in traditional metrics as well as using the "eye" test. Finally, a note on clarity: several figures, tables, and their captions are ambiguous, and the authors are encouraged to consider rewriting several pieces. To give some examples: Table 3 is ambiguous; what exactly is being measured, and can the authors provide precise wall-clock runtimes?  Where do the approximate runtimes (e.g "~35 minutes") reported in Figure 5 come from --- are these average timing results of multiple runs, or their median? Another example is Figure 11; it is unclear what the reader expected to visualize from the inset squares.

I encourage the authors of the paper to prepare future revisions of the manuscript while keeping these points in mind.

**Audience:**

Yes

**Audience Explanation:**

The paper addresses a topic in text-to-3D generative models which is a currently-active area of computer vision and machine learning research.

**Claims And Evidence:**

No

**Claims Explanation:**

This paper proposes MVGaussian, a new way to generate 3D content from textual input. This is a (very) well-studied problem in the recent literature in 3D vision, and the main novelty is the careful mix of a few algorithmic components: multi-view information (to solve the so-called Janus problem), a specific regularizer for 3D Gaussian splats and a densification strategy to align Gaussians close to the surface of the object.

While the paper is interesting, certain claims in the manuscript lack clarity and/or sufficient evidence.

Within the framework of multi-view guidance for text-to-3D generation (which has already been explored in the literature), the paper's conceptual innovations appear to be two-fold: an extra regularization term for surface alignment that yields better integrity/fidelity of the generated models, and a substantial training efficiency gain over recent 3DGS methods. Both claims need either additional evidence or further refinement. First, ablations on the extra term  do not improve qualitative results (and for this problem, quantitative metrics are challenging). Second, training efficiency gains are not clearly articulated, and the speedups are reported in an imprecise manner under ambiguous experimental conditions. See also below the text box on additional comments.

**Resubmission Of Major Revision:**

The authors may consider submitting a major revision at a later time.